# A conserved Pol II elongator SPT6L mediates Pol V transcription to regulate RNA-directed DNA methylation in *Arabidopsis*

Yujuan Liu[1,2,3], Jie Shu[2,4], Zhi Zhang[1,2,3], Ning Ding[2,5], Jinyuan Liu[1,2,3], Jun Liu[4], Yuhai Cui ® [6,7], Changhu Wang[1,2,3] ✉ & Chen Chen ® [1,2,3] ✉

In plants, the plant-specific RNA polymerase V (Pol V) transcripts non-coding RNAs and provides a docking platform for the association of accessory proteins in the RNA-directed DNA methylation (RdDM) pathway. Various components have been uncovered that are involved in the process of DNA methylation, but it is still not clear how the transcription of Pol V is regulated. Here, we report that the conserved RNA polymerase II (Pol II) elongator, SPT6L, binds to thousands of intergenic regions in a Pol II-independent manner. The intergenic enrichment of SPT6L, interestingly, co-occupies with the largest subunit of Pol V (NRPE1) and mutation of SPT6L leads to the reduction of DNA methylation but not Pol V enrichment. Furthermore, the association of SPT6L at Pol V loci is dependent on the Pol V associated factor, SPT5L, rather than the presence of Pol V, and the interaction between SPT6L and NRPE1 is compromised in *spt5l*. Finally, Pol V RIP-seq reveals that SPT6L is required to maintain the amount and length of Pol V transcripts. Our findings thus uncover the critical role of a Pol II conserved elongator in Pol V mediated DNA methylation and transcription, and shed light on the mutual regulation between Pol V and II in plants.

In eukaryotic cells, transcription elongation is a dynamic and highly regulated process, in which a variety of functionally distinct transcript elongation factors are involved in Pol II progression[1,2]. Among them, the conserved elongator, SPT6, is recruited by the phosphorylated Pol II[3] and involved in the enhancement of elongation rate[4–6], repression of intragenic initiation[7,8], and transcription termination[5] in yeast and animal cells. In plants, the functional homolog of SPT6, SPT6-like (SPT6L), interacts with phosphorylated Pol II and plays conserved roles in Pol II progression[9]. The mutation of SPT6L causes pleiotropic defects in embryogensis[10] and post-germination stages[9]. Recently, it

was found that SPT6L was able to recruit chromatin remodelers SWI2/SNF2 at transcription start sites (TSS) in a Pol II-independent manner[11], indicating a potential role of SPT6L in transcription initiation/early elongation in plants.

Different from animal and yeast cells, in plants, two plant-specific RNA polymerases (Pol IV and V) have evolved and they play essential roles in the establishment and maintenance of DNA methylation through the RdDM pathway[12,13]. In general, the canonical RdDM pathway is composed of two parts: the production of 24-nt siRNA and the establishment of DNA methylation[13]. The production of 24-nt siRNA is

[1]State Key Laboratory of Plant Diversity and Specialty Crops, South China Botanical Garden, Chinese Academy of Sciences, Guangzhou, Guangdong 510650, China. [2]Key Laboratory of South China Agricultural Plant Molecular Analysis and Genetic Improvement & Guangdong Provincial Key Laboratory of Applied Botany, South China Botanical Garden, Chinese Academy of Sciences, Guangzhou, Guangdong 510650, China. [3]University of the Chinese Academy of Sciences, Beijing 100049, China. [4]Guangdong Academy of Agricultural Sciences, Guangzhou, Guangdong 510640, China. [5]MOE Key Laboratory of Cell Activities and Stress Adaptations, School of Life Sciences, Lanzhou University, Lanzhou 730000, China. [6]Agriculture and Agri-Food Canada, London Research and Development Centre, London, ON N5V 4T3, Canada. [7]Department of Biology, Western University, London, ON N6A 5B7, Canada. ✉e-mail: wangchangh@scib.ac.cn; chenchen101@scbg.ac.cn

accomplished by Pol IV's transcription, RNA-DEPENDENT RNA POLY-MERASE2 (RDR2)'s generation of double-stranded RNA, and DICER-LIKE PROTEINs (DCLs) dependent cleavage[13]. In the second part, Pol V transcripts serve as a docking platform to recruit AGOs-siRNA complex and other accessory proteins to establish DNA methylation. The RdDM pathway is a self-reinforcing loop[14] and the reduced siRNA and DNA methylation negatively affect the transcription of Pol V[12,13,15].

Unlike the processive transcription of Pol II, the estimated transcripts of Pol IV and V are short (30 to 40 nt[16,17] and around 200 nt[18], respectively) in length. Previous in vitro data indicates that both Pol IV and V can transcript on bipartite DNA-RNA templates and only Pol IV maintains the transcription ability on tripartite template[19], suggesting Pol V prefers to single strand DNA as template and is lack of ability to displace the non-template DNA. Recent structural data also shows that the conserved tyrosine residue of NRP(D/E)2, the second subunit of Pol IV and V, can stall transcription and enhance backtracking by interacting with non-template DNA strand[20]. In addition, the lack of surfaces to recuit Pol II transcription factors such as TFIIB and TFIIS[17,20] also suggested that Pol IV and V acts via a distinct regulatory mechanism compared to Pol II. Although the above in vitro and structural data revealed the nature of Pol IV and V in transcription, it is still not clear how the transcription of Pol IV and V are regulated in vivo and what factors are involved in the above process to distinguish the different transcription behaviors of Pol IV and V.

In this work, we found that the conserved elongator, SPT6L, was enriched at thousands of intergenic regions in a Pol II-independent manner. Interestingly, NRPE1, the largest subunit of Pol V, was also highly enriched in those regions. Mutation of SPT6L led to the reduction of DNA methylation but not the association between Pol V and chromatin. Further analyses showed that the associated protein, SPT5L, rather than the presence of Pol V is indispensable for the intergenic enrichment of SPT6L, and the interaction between SPT6L and NRPE1 was compromised in *spt5l*. Finally, NRPE1 RIP-seq indicated that SPT6L is required to maintain the amount and length of Pol V transcripts. Taken together, our work revealed a Pol II and V shared component and its roles in the maintenance and promotion of DNA methylation and Pol V transcripts, respectively.

## Results

### SPT6L associates and co-occupies with the Pol V complex

Our previous data revealed that the elongation factor, SPT6L, associates with Pol II and plays roles in transcription initiation and elongation[9,11]. When browsing the occupancy signals of SPT6L, we have noticed the intergenic enrichment of SPT6L (Fig. 1a). To examine the intergenic enrichment of SPT6L in detail, we reanalyzed previous ChIP-seq data[11] and identified 2325 intergenic peaks across the genome (Supplementary Data 1). Within those regions, interestingly, we only detected the enrichment of SPT6L but not Pol II (Supplementary Fig. 1a), which recruited SPT6L during the transcription of protein-coding genes[9]. We further analyzed the overlapping of the intergenic peaks with different genome features and found that transposons were highly enriched within those peaks (Supplementary Fig. 1b), suggesting a potential link between SPT6L and the regulation of transposable elements (TEs).

The unexpected enrichment of SPT6L at transposons (Supplementary Fig. 1b) and its conserved roles in DNA-dependent RNA polymerases[9,21,22] prompted us to examine the potential interplay between SPT6L and Pol IV/V, which play major roles in the silencing of transposons[12]. We profiled the published ChIP-seq signals of NRPE1[23], the largest subunits of Pol V, at SPT6L binding sites and found highly enriched NRPE1 signals at the intergenic peaks of SPT6L (Fig. 1a and Supplementary Fig. 1a). By comparing the binding peaks of SPT6L and NRPE1, we found 6008 NRPE1 peaks overlapped with SPT6L peaks (Supplementary Data 1) and a stronger binding strength of NRPE1 at the overlapped peaks than that at NRPE1-only peaks (Supplementary

Fig. 1c). We further compared the frequency of TE within NRPE1 peaks and observed a higher amount of TE in NRPE1-SPT6L overlapped peaks than that in NRPE1-only peaks (Supplementary Fig. 1d). To clarify the types of TE associated to NRPE1 and NRPE1-SPT6L peaks, we found that the NRPE1-bound TEs were more abundant in Helitron and SINE, but less enriched in Gypsy (Supplementary Fig. 1e). The compositions of TE within NRPE1 and NRPE1-SPT6L peaks were similar (Supplementary Fig. 1e), indicating that there is no preference of NRPE1-SPT6L peaks in different TE types.

As the published ChIP-seq data of NRPE1 was sourced from inflorescence[23], we decided to generate GFP-tagged NRPE1 transgenic lines and profiled the genome-wide occupancy of NRPE1 with the same plant tissues as those used for generating the SPT6L's data. Firstly, we confirmed the normal function of NRPE1-GFP (*nrpe1-11 pNRPE1:NRPE1-GFP*, hereafter all *nrpe1-11* were named *nrpe1*) by examining the nuclear-localized GFP signals (Supplementary Fig. 2a) and recovered DNA methylation at selected RdDM loci (Supplementary Fig. 2b). And then, we profiled the genome-wide occupancy of NRPE1 and identified 7809 confident peaks (Irreproducible Discovery Rate, IDR < 0.01, Supplementary Data 1) across two biological replicates. Those peaks were highly overlapped with the published NRPE1 binding peaks (Supplementary Fig. 2c). By comparing the binding signals of NRPE1 and SPT6L, consistently, we detected similar co-binding signals between SPT6L and NRPE1 at SPT6L intergenic peaks, where were lack of Pol II signals (Fig. 1b). Finally, to define SPT6L and NRPE1 co-bound genomic regions in a reliable and unbiased way, we divided the genome into 200 bp bins and used a Hidden Markov Model (HMM) to identify bins with enrichment of SPT6L and NRPE1. When the normalized SPT6L, NRPE1, and Pol II ChIP-seq data combined from two biological replicates were analyzed this way, the genome could be split into six groups (Supplementary Data 2) and, importantly, the NRPE1-only (G2, weak SPT6L signals) and NRPE1-SPT6L shared (G3, strong SPT6L signals) bins (Fig. 1c and Supplementary Fig. 2d) were clearly distinguished. Consistently, the NRPE1-SPT6L shared regions contained more TE loci than the NRPE1-only ones (Supplementary Fig. 2e). Although the NRPE1 signals were generally enriched upstream of transcription start sites (TSS), the NRPE1-SPT6L shared and NRPE1-only bins were, interestingly, distinguished around 400 (distal) and 200 (proximal) bp upstream of TSS, respectively (Fig. 1d), suggesting that these two different binding patterns of NRPE1 may have distinct roles.

To examine the potential association between SPT6L and Pol V complex, we performed yeast-two-hybrid assays between SPT6L and multiple subunits, which were distinct between Pol II and V (NRPE1, NRP(D/E)2, NRP(D/E)4, NRP5, and NRPE7)[24]. As shown in Supplementary Fig. 2f, g, while no interaction was found between SPT6L and NRPE1, we find SPT6L can directly interact to NRP(D/E)4 in yeast. And then, we further confirmed the observed interactions among SPT6L and several subunits of Pol V by co-immunoprecipitation (Co-IP) with stable transgenic lines (Fig. 1e; Supplementary Fig. 2a, h). Consistently, we were also able to detect the associations between SPT6L and multiple subunits of Pol V in vivo (Fig. 1e), indicating that SPT6L can form protein complex with Pol V in *planta*. Altogether, the above results indicate that SPT6L probably collaborates with Pol V to mediate the silencing of transposons in plants.

### Pol V is required for the intergenic recruitment of SPT6L

The interaction and genomic co-occupancy between SPT6L and Pol V prompted us to examine the mutual dependency of their genome recruitment. Therefore, we profiled and compared the genome-wide occupancy of NRPE1 in WT (*nrpe1 pNRPE1:NRPE1-GFP*) and *spt6l* (*nrpe1 spt6l pNRPE1:NRPE1-GFP*) backgrounds and found that the overall enrichment of NRPE1 in *spt6l* was unchanged in both G3 (NRPE1-SPT6L shared) and G2 (NRPE1-only) regions (Fig. 2a). The following ChIP-qPCR at selected RdDM loci also confirmed the general unchanged pattern of NRPE1 occupancy in *spt6l*

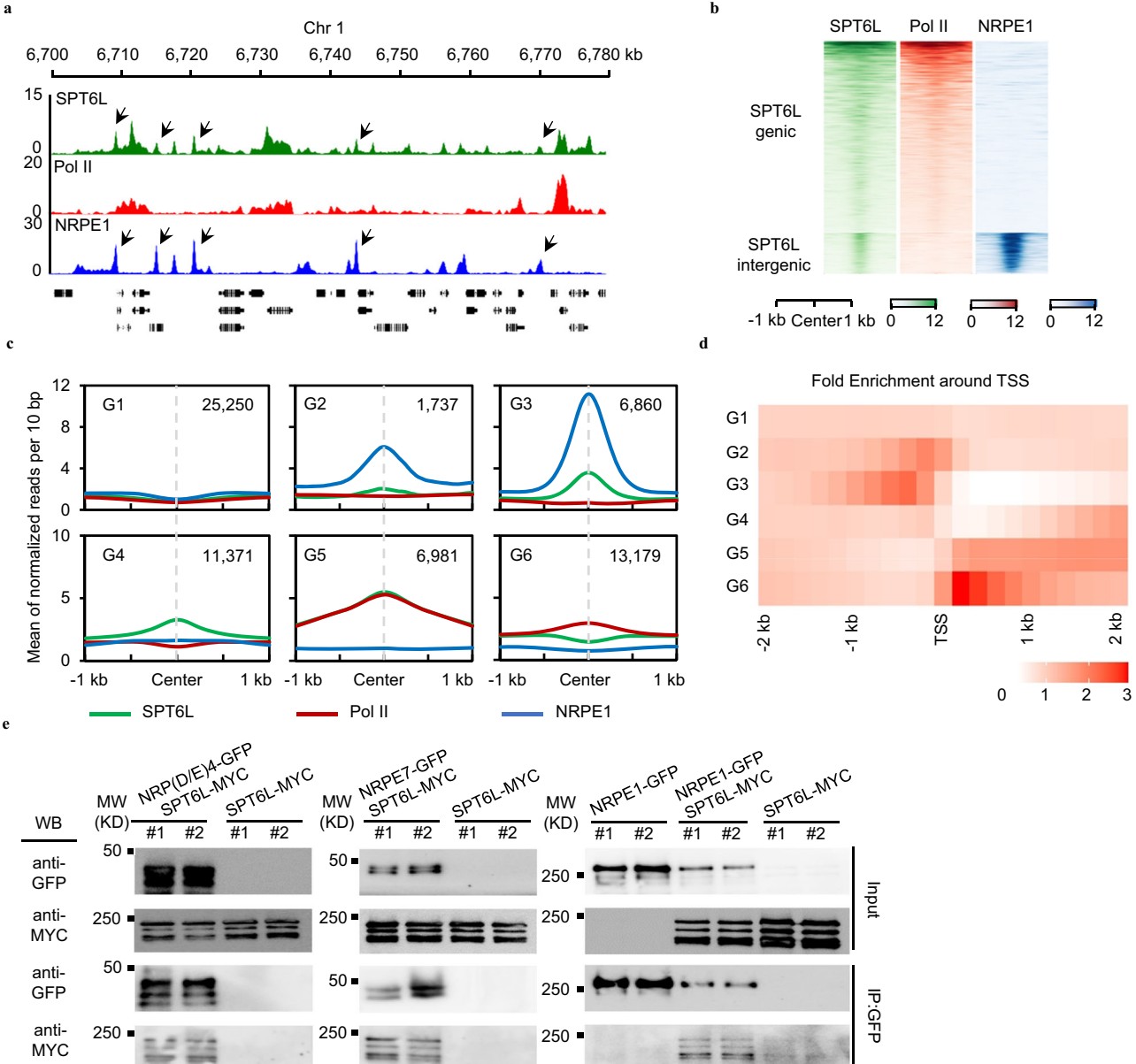

**Fig. 1 | SPT6L co-occupies and interacts with NRPE1 in Arabidopsis. a** Genome tracks display SPT6L, Pol II, and NRPE1 ChIP-seq signals on chromosome 1 (Chr1: 6700 to 6780 kb). The ChIP signal of each sample was averaged over two biological replicates. The y-axis values indicate the mean of normalized reads per 10 bp. The black arrows pointed to co-binding peaks between SPT6L and NRPE1. **b** Heatmaps of SPT6L, Pol II, and NRPE1 ChIP signals around peak center of all SPT6L peaks. The SPT6L peaks were clustered into two groups (genic and intergenic). The plotted regions are upstream and downstream 1 kb of peak center. **c** Binding profiles of SPT6L, Pol II, and NRPE1 at characterized six genomic groups. The six groups of regions were clustered according to the solo/double enrichment among the three proteins (see Supplementary Fig. 2d). The ChIP signal of each sample was averaged over two biological replicates. The plotted regions were 2 kb around the center of regions (upstream and downstream 1 kb, respectively). The y-axis value indicates the relative mean of normalized reads (1× sequencing depth normalization) per 10 bp non-overlapping bins. The number of regions in each group (G1 to G6) was indicated in the graph. **d** Heatmaps of the distance between enriched regions in each state and transcription start sites (TSS). Each rectangle represents 200 bp. **e** Co-immunoprecipitation (Co-IP) examined the interaction between SPT6L and core subunits of Pol V complex. Immunoprecipitation (IP) and Western blot (WB) were performed using specified antibodies (IP: anti-GFP, KTSM1301; WB: anti-Myc, ab9106, anti-GFP, Yeasen, 31002ES60). Data from two biological replicates were shown.

(Supplementary Fig. 3a). We further assessed the protein stability of NRPE1 in *spt6l* and detected a comparable protein level of NRPE1 in both WT and *spt6l* (Supplementary Fig. 3b). By comparing the genome-wide profiles of NRPE1 in WT and *spt6l*, we found that the ChIP reads of NRPE1 in both genotypes were highly correlated (Fig. 2b). These results indicate that SPT6L is dispensable for the enrichment of NRPE1. Next, we examined the dependency of SPT6L on NRPE1 by ChIP-seq in WT and *nrpe1* mutant backgrounds. As shown in Fig. 2c, the mutation of *NRPE1* dramatically reduced the occupancy of SPT6L at G3 (NRPE1-SPT6L shared) regions but not

other SPT6L enriched regions (G4 and G5). These results were then confirmed by ChIP-qPCR at selected loci (Supplementary Fig. 3c) and immunoblotting showed a comparable protein level of SPT6L in both WT and *nrpe1*(Supplementary Fig. 3d). This result indicates that NRPE1 is required for the intergenic enrichment of SPT6L in plants.

Although the dependency of SPT6L on NRPE1 shown above explained the co-occupancy of SPT6L and NRPE1 at the G3 regions, it is still not clear why there were less enriched SPT6L signals at the G2 regions, which showed a moderate NRPE1 signal (Fig. 1c). We noticed

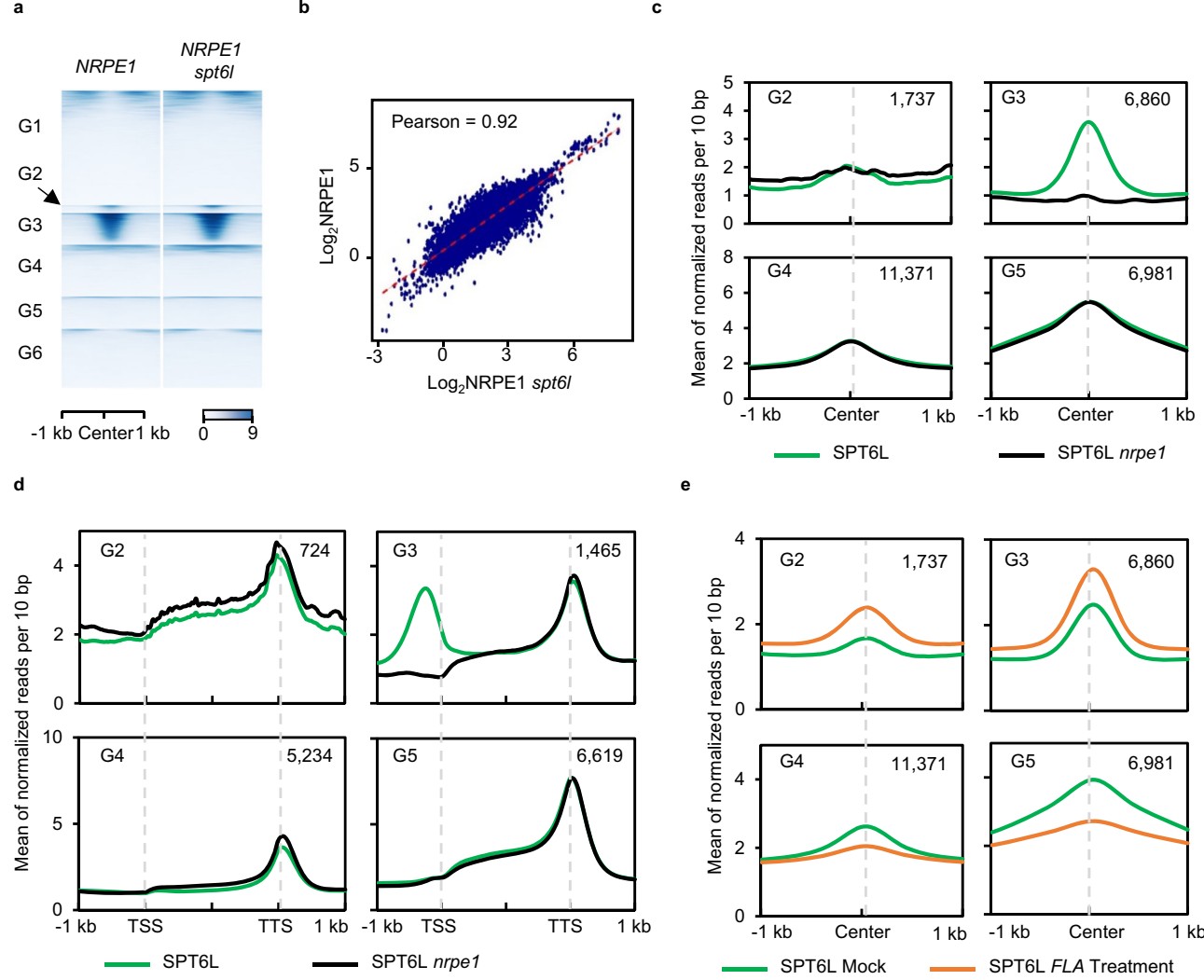

**Fig. 2 | NRPE1 is required for the intergenic enrichment of SPT6L. a** Heatmaps of NRPE1 ChIP signals in *nrpe1 NRPE1-GFP* (*NRPE1*) and *nrpe1 spt6l NRPE1-GFP* (*NRPE1 spt6l*) background. The plotted regions were similar to Fig. 1c and the strength of ChIP signals at 1 kb around region centers was shown. **b** Scatterplot of NRPE1 ChIP signals in *NRPE1* and *NRPE1 spt6l* at Pol V peaks. ChIP signals (log₂ values) in *NRPE1* (y-axis) and *NRPE1 spt6l* (x-axis) were plotted. **c** Binding profiles of SPT6L in WT and *nrpe1* at four previously defined genomic groups. The ChIP signal of each sample was averaged over two biological replicates. The regions were plotted as indicated in Fig. 1c. **d** Binding profiles of SPT6L occupancy in WT and *nrpe1* at the nearest downstream TSS of four genomic states. Plotting regions were scaled to the same length as follows: 5′ ends (−1.0 kb to TSS) and 3′ ends (transcription termination site [TTS] to downstream 1.0 kb) were not scaled, and the gene bodies were scaled to 2 kb. The y-axis was plotted as described in Fig. 1c. The number of genes was indicated (*n*). **e** Binding profiles of SPT6L occupancy at the four genomic states after 1 h mock and Flavopiridol (FLA) treatment. The regions were plotted as indicated in Fig. 1c.

that the G2 regions were enriched around 200 bp upstream of TSS, which is closer than that of G3 regions to TSS (Fig. 1d). As there is strong association of SPT6L with phosphorylated Pol II around TSS[9], we assumed that the weak signals of SPT6L at the G2 regions may result from the local competition between Pol II and V in the recruitment of SPT6L. To test this assumption, firstly, we identified the nearest downstream TSS of each previously defined genomic group (G2 to G5) (Supplementary Data 3) and compared the SPT6L ChIP signals around those TSS. As shown in Fig. 2d, the mutation of *NRPE1* led to a slight but clear increase of SPT6L occupancy only at the downstream genes of the G2 regions, suggesting that the presence of upstream NRPE1 in the G2 regions may either trap SPT6L or directly inhibit transcription. And then, we compared the SPT6L ChIP signals[9] at four different groups after treating with a P-TEFb inhibitor (Flavopiridol, FLA), which decreases the phosphorylation levels of Pol II and disrupts its interaction with SPT6L[9,11]. Indeed, the application of the inhibitor reduced the occupancy of SPT6L at genic regions (G4 and G5) (Fig. 2e and Supplementary Fig. 2e). Meanwhile, we also detected increased

occupancies of SPT6L at both G2 and G3 regions (Fig. 2e), indicating that the dissociation of SPT6L to Pol II can increase the occupancy of SPT6L at NRPE1 binding sites. Altogether, the above results suggested that Pol II and V may compete to recruit SPT6L to facilitate its transcription in plants.

**Mutation of SPT6L causes the reduction of DNA methylation**

The essential role of Pol V in RdDM and the intergenic enrichment of SPT6L led us to further examine the potential effects of SPT6L on DNA methylation in plants. We first performed Chop-PCR to examine the DNA methylation at several known RdDM loci. As shown in Fig. 3a, the DNA methylation levels were reduced but not eliminated at the *SN1*, *IGN5*, and *IGN23* loci in *spt6l*. And then, to assess the generality of these findings, we performed whole-genome bisulfite sequencing analyses (BS-seq) and identified 4099 differentially methylated regions (DMRs) in *spt6l* (Supplementary Data 3). Most of the DMRs (3,681 out of 4,099) were hypomethylated. Similar to what was found in *nrpe1*, the identified hypo DMRs showed

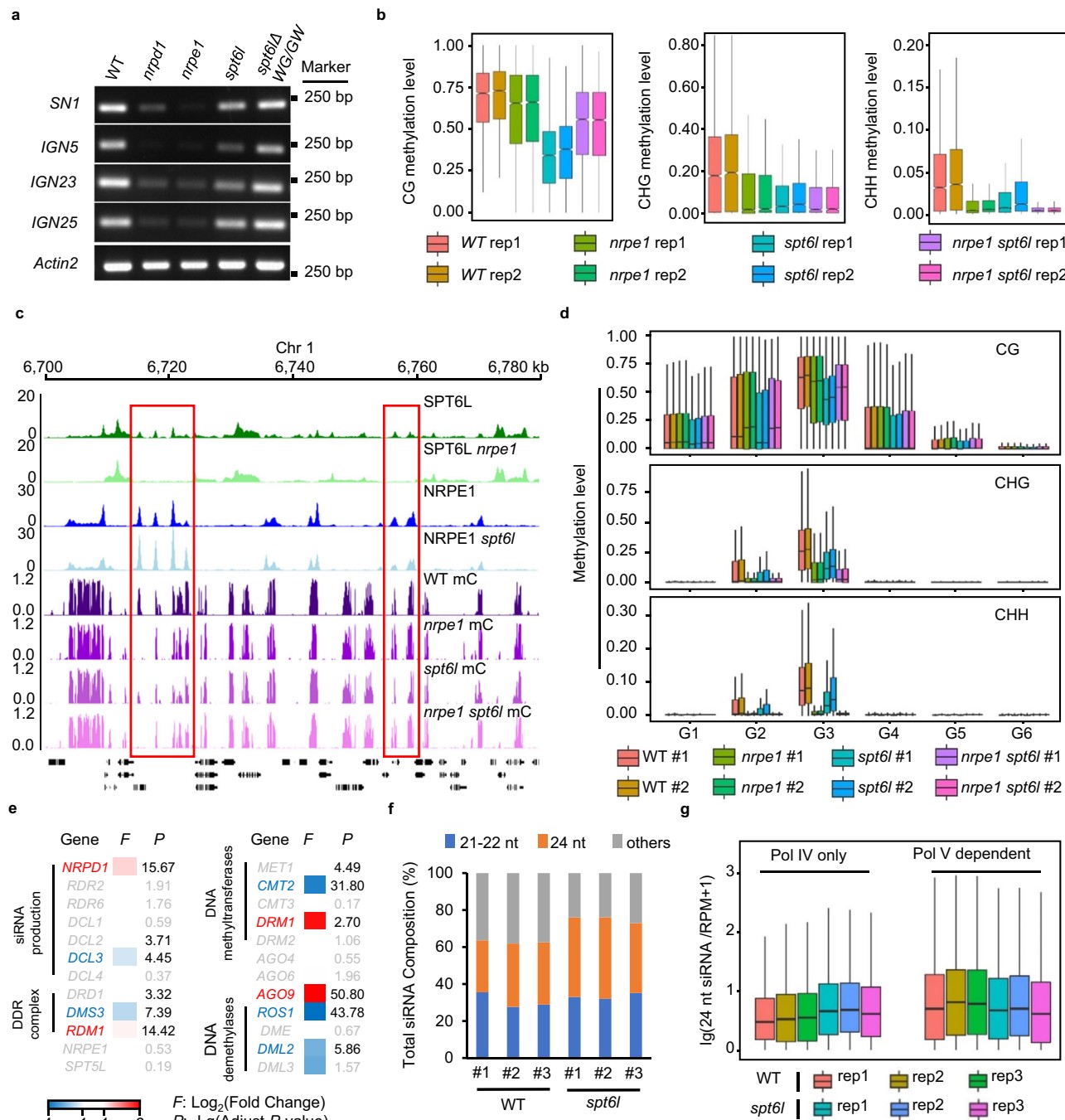

**Fig. 3 | SPT6L is involved in the regulation of DNA methylation. a** Chop-PCR analysis of DNA methylation at selected loci performed by digestion with *Hae* III restriction endonuclease. Digested genomic DNA was amplified by PCR. Sequences lacking *Hae* III (*Actin2*) were used as loading controls. **b** Boxplots of DNA methylation level at three contexts (CG, CHG, and CHH) among different samples. The plotted regions were *spt6l* DMRs and Data from two biological replicates were shown. **c** Genome tracks display the ChIP-seq and BS-seq signals on chromosome 1. The ChIP signals included SPT6L and NRPE1 in WT/*nrpe1* and WT/*spt6l* background, respectively. The BS-seq signals showed total methylation levels in *WT*, *nrpe1*, *spt6l*, and *nrpe1 spt6l*. Each sample was averaged over two biological replicates. The red rectangle highlighted SPT6L and NRPE1 co-targeted loci with changed DNA methylation levels. **d** Boxplots of DNA methylation level at three contexts (CG, CHG, and CHH)

among different samples. The plotted regions were previously defined six genomic groups and Data from two biological replicates were shown. **e** Heatmap showed the expression of RdDM pathway-related genes by RNA-seq. The threshold to define differentially expressed genes is more than 2-fold expression change (*F*) and adjust *p*-value (*P*)less than 0.01. The up and down regulated genes were indicated as red and blue, respectively. The genes with either less than 2-fold change or *p*-value larger than 0.01 were labeled as gray. Three biological replicates were included. **f** Stacked bar graph showed the proportion of different size of small RNA in WT and *spt6l*. Data from three biological replicates were shown. **g** Boxplots showed the amount of 24-nt siRNA in WT and *spt6l* at Pol IV and V-dependent regions. Data from three biological replicates were shown. The center line of boxplot, median; box limits, upper and lower quartiles; whiskers, 1.5 × interquartile range.

hypomethylation at the CHG and CHH contexts (hypo mCHG and mCHH) compared to WT (Fig. 3b). We then performed a BS-seq in the *nrpe1 spt6l* double mutant and revealed a similar hypo mCHG and mCHH to that in *nrpe1* (Fig. 3b), indicating that the mCHG/mCHH in

hypo DMRs of *spt6l* mainly NRPE1-dependent, and SPT6L may be involved in mCHG/mCHH through NRPE1. Interestingly, the mutation of SPT6L also caused a dramatical reduction of DNA methylation at the CG context (mCG), which was only slightly reduced in

*nrpe1*(Fig. 3b), indicating that SPT6L may also regulate DNA methylation in NRPE1 independent manner. To examine whether SPT6L directly contributed to the DNA methylation at the hypo DMRs in *spt6l*, we integrated our ChIP-seq and BS-seq data and found that the decreased methylation, even in the CG context, was mainly detected at G2 and G3 regions (Fig. 3c, d), suggesting that SPT6L likely contributes to DNA methylation mainly through NRPE1 mediated DNA methylation.

As SPT6L interacts with Pol II and plays an essential role in transcription[9,21], the reduced DNA methylation may result from the mis-expressed genes encoding DNA methyltransferases and demethylases in *spt6l*. Thus, to assess the possibility, we performed RNA-seq assay with three biological replicates and compared the expression of DNA methyltransferases and demethylases in WT and *spt6l*. In total, we have detected more than 12,000 differentially expressed genes in *spt6l* with DEseq2 package (Supplementary Fig. 4a and Data 4, adjust $P < 0.01$, |Fold Change| ≥ 2). Within the five major DNA methyltransferase genes (*MET1, CMT2, CMT3, DRM1*, and *DRM2*)[25], we found that only the expression of *CMT2* was significantly downregulated in *spt6l* (Fig. 3e). Except for *CMT2*, the *DRM1* and the other three genes encoding DNA methyltransferase showed increased and unchanged expression in *spt6l* mutant, respectively (Fig. 3e). Meanwhile, we also examined the expression of four major DNA demethylase genes (*ROS1, DME, DML2*, and *DML3*) and detected a significantly decrease of *ROS1* and *DML2* in *spt6l* (Fig. 3e). To further estimate the potential effects of malfunctioned transcription on DNA methylation, we took advantage of the published DNA methylation datasets[26] and compared the methylation levels at *spt6l* DMRs in mutants of five methyltransferases, Pol IV (*nrpd1*), and Pol II (*nrpb2*). As shown in Supplementary Fig. 4b, the changed DNA methylation patterns at the three contexts in *spt6l* were distinct from that in *cmt2*, which caused changed DNA methylation mainly at CHG/CHH. Altogether, the above results suggest that the reduced DNA methylation in *spt6l* is less likely resulted from the misregulation of DNA methyltransferases and demethylases.

Other than DNA methyltransferases, the biogenesis of small interfering RNA (siRNA) also plays an essential role in both the canonical and non-canonical RdDM pathways in plants[13]. In addition, the previously identified interaction between SPT6L and NRP(D/E)4 (Supplementary Fig. 2f, g) also raised the potential linkage of SPT6L to siRNA production via Pol IV complex. Thus, we examined the expression of multiple components related to the production of siRNA and found the expressions of *NRPD1* and *DCL3* were altered in *spt6l* (Fig. 3e). To directly estimate the potential effect of SPT6L on the production of siRNA, we performed small RNA deep sequencing in WT and *spt6l* for comparison. Even though some of the 21-22 and 24-nt siRNA produced by Pol II[27], interestingly, we did not detect any dramatic change in the compositions of the 21-22 and 24-nt siRNAs in *spt6l* (Fig. 3f). Previously, the 24-nt siRNAs have been clustered into upstream (siRNAs dependent on Pol IV only) and downstream siRNAs (siRNAs dependent on both Pol IV and Pol V)[28]. The upstream siRNAs are affected only in mutants defective in upstream RdDM components, such as *nrpd1*, whereas the downstream siRNAs are affected in the mutants of both Pol IV and V-related components. To carefully assess the role of SPT6L in the biogenesis of siRNA, we compared the amounts of 24-nt siRNA in the above two clusters between WT and *spt6l*. As shown in Fig. 3g, a slightly decreased amount of the 24-nt siRNA was found in Pol V-dependent regions, although the total composition of 24-nt siRNA was not reduced in *spt6l* (Fig.3f). Meanwhile, we also detected an unchanged or even slightly increased siRNA in Pol IV only regions (Fig. 3g), which may result from the up-regulation of *NRPD1* in *spt6l* (Fig. 3e). These results indicated that SPT6L is not involved in the production of siRNA and the reduced DNA methylation in *spt6l* unlikely results from the alternation of siRNA.

## The WG/GW repeat of SPT6L is dispensable for RdDM

The C-terminals of both NRPE1 and SPT5L contain a WG/GW repeat, which is essential for the AGO4 recruitment and DNA methylation at RdDM loci[29]. Interestingly, a WG/GW repeat was also found at the C-terminal of SPT6L, which was computationally scored in the top 3 of Argonaute (AGO) interacting proteins[30]. To examine whether the WG/GW repeat of SPT6L contributes to the enrichment of SPT6L at RdDM loci and DNA methylation, we generated a WG/GW deleted construct and introduced it into *spt6l*[+/-]. As shown in Fig. 4a, the truncated SPT6L was able to rescue the developmental defects of *spt6l*. The transgenic line of *spt6l SPT6LΔWG/GW-GFP* was further validated by confirming its nuclear localization signals and comparable protein levels to that of SPT6L (Supplementary Fig. 5a, b). And then, the genome-wide occupancy of SPT6LΔWG/GW-GFP was profiled and a similar binding pattern and high correlation were revealed between SPT6LΔWG/GW and SPT6L (Fig. 4b, c), suggesting that the WG/GW repeat is not required for the transcriptional function of SPT6L under normal condition. Especially, the occupancy of SPT6LΔWG/GW at the SPT6L intergenic regions was comparable to that in SPT6L (Fig. 4b), indicating that the WG/GW repeat is also dispensable for the association of SPT6L to RdDM loci. To examine the role of SPT6L-WG/GW in DNA methylation, we performed Chop-PCR and found that the introduction of SPT6LΔWG/GW was able to recover the reduced DNA methylation at selected RdDM loci (Fig. 3a). We then performed BS-seq and detected similar genome-wide DNA methylation levels between WT and *spt6l SPT6LΔWG/GW* (Fig. 4d, e). Altogether, these results indicate that the WG/GW repeat of SPT6L, unlike that of NRPE1 and SPT5L, is dispensable for SPT6L's genomic recruitment and role in DNA methylation at the RdDM loci.

## SPT5L is required for the recruitment of SPT6L to RdDM loci

In the RdDM pathway, following the recruitment of Pol V, multiple proteins are bound to Pol V/Pol V transcripts and mediate the DNA methylation[13]. To further clarify whether the enrichment of SPT6L at intergenic loci is dependent on Pol V or the downstream events, we examined the genome-wide occupancy of SPT6L in *spt5l*, which impairs the slicing features of Pol V transcripts but not the enrichment of Pol V[31–33]. Interestingly, the occupancies of SPT6L at the NRPE1-related regions (G2 and G3) but not the other SPT6L enriched regions (G4 and G5) were dramatically reduced in *spt5l* (Fig. 5a, b), indicating that SPT5L is required for the intergenic recruitment of SPT6L. We confirmed the results at selected genomic loci by ChIP-qPCR (Supplementary Fig. 6a), and our immunoblotting assay showed that the altered enrichment of SPT6L did not result from any potential changes of protein stability in *spt5l* (Supplementary Fig. 6b). To confirm the potential effects of *spt5l* on the binding of NRPE1, we examined the enrichment of NRPE1 at selected loci in *spt5l* by ChIP-qPCR. Consistent with previous results[33], the binding of NRPE1 was generally unchanged in *spt5l* at selected loci (Fig. 5c). These data indicated that the presence of Pol V alone is insufficient to determine the binding of SPT6L. To further examine the essential role of SPT5L in the recruitment of SPT6L, we firstly examined the interaction between them by yeast-two-hybrid assays. As shown in Supplementary 6c, an interaction between SPT5L and SPT6L was detected in yeast and the further truncations of SPT5L revealed the N-terminal of SPT5L played a major role in its interaction with SPT6L. To confirm the interaction in vivo, stable transgenic plants containing *pSPT5L:SPT5L-GFP* and *pSPT6L:SPT6L-MYC* were generated and the interaction between SPT5L and SPT6L was confirmed by Co-IP assay (Fig. 5d). Finally, by knocking out SPT5L, we detected a compromised interaction between SPT6L and NRPE1 (Fig. 5e), indicating that SPT5L is indispensable for the recruitment of SPT6L into the Pol V complex.

In addition, we also examined the occupancy of SPT6L in the mutants of *drm1 drm2*, and *nrpd1*. The former plays a role in the downstream of Pol V and catalyzes DNA methylation[14]. The latter

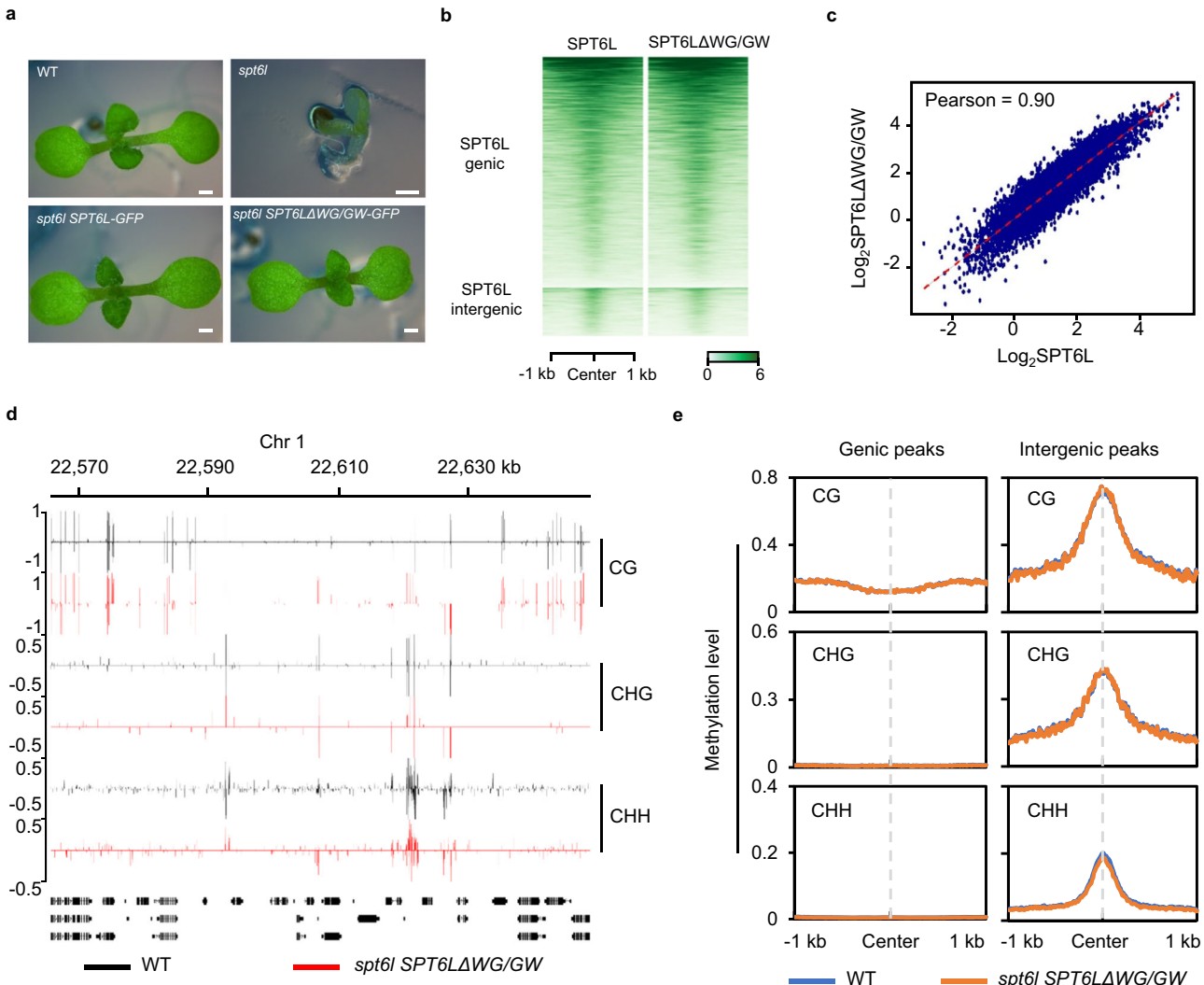

**Fig. 4 | The WG/GW repeat of SPT6L is dispensable for its intergenic enrichment and DNA methylation. a** The morphological phenotypes of 7-day old *WT*, *spt6l*, *spt6l pSPT6L:SPT6L-GFP*, and *spt6l pSPT6L:SPT6LΔWG/GW* seedlings. Bar = 0.5 mm. **b** Heatmaps of SPT6L and SPT6LΔWG/GW ChIP signals at SPT6L binding peaks. The plotted regions were similar to Fig. 1b and the strength of ChIP signals at 1 kb around of region centers were shown. **c** Scatterplot of SPT6L and SPT6LΔWG/GW ChIP signals in genomic binds. The Arabidopsis genome was divided into 100 bp

length bins. ChIP signals (log₂ values) in SPT6LΔWG/GW (y-axis) and SPT6L (x-axis) were plotted. **d** Genome tracks display the DNA methylation signals (in CG, CHG, and CHH contexts) of *WT* and *spt6l SPT6LΔWG/GW* on part of chromosome 5. Data from one biological replicate was shown. **e** Plots of DNA methylation levels in *WT* and *spt6l SPT6LΔWG/GW* within the genic and intergenic peaks of SPT6L. The plots displayed DNA methylation in CG, CHG, and CHH context. Data from one (*spt6l SPT6LΔWG/GW*) or two (*WT*) biological replicate(s) were shown.

encodes the largest subunit of Pol IV and determines the production of 24-nt siRNA in plants[34]. As shown in Fig. 5f, the occupancies of SPT6L in *drm1 drm2*, and *nrpd1* are significantly reduced at some but not all selected loci, suggesting that Pol IV and DRM1/DRM2 may affect the intergenic enrichment of SPT6L in a loci-specific manner. Immuno-blotting confirmed that the changed occupancies of SPT6L in *drm1 drm2* and *nrpd1* did not result from protein stability (Supplementary Fig. 6d). Previously, it was reported that the occupancy of Pol V in *nrpd1* and *drm1 drm2* were slightly reduced[23,32,35]. Thus, the loci-specific reduction of SPT6L in *nrpd1* and *drm1 drm2* may result from the decreased occupancy of Pol V.

## SPT6L is required for Pol V transcription

In eukaryotic cells, SPT6L(SPT6) plays an essential role in Pol II elongation[5,9]. The association of SPT6L with Pol V led us to examine the potential functions of SPT6L in Pol V transcription. For that, we performed RIP-seq in *nrpe1*, *NRPE1-GFP nrpe1*(*NRPE1-GFP*), and *NRPE1-GFP nrpe1 spt6l* (*spt6l NRPE1-GFP*) by using a GFP antibody. To minimize the effect of mechanical force on Pol V transcripts, we replaced the

sonication step with DNase I treatment in the original IPARE protocol[31]. As shown in Fig. 6a–c, the Pol V transcripts can be detected in NRPE1 peaks and the amount of Pol V transcripts within NRPE1 peaks was significantly reduced in *spt6l*, indicating that SPT6L is required for promoting Pol V transcription. Furthermore, by comparing the RIP-seq reads, we also found a significant reduction of the length of RIP-seq reads in *spt6l* (Fig. 6d), suggesting that SPT6L may play a role in Pol V elongation. Other than the quantity and length of RIP-seq reads, Pol V transcripts can be sliced by AGO4 and create a strong signature for uridine (U) at position 10 (U-10)[32,36]. To examine whether the slicing feature was also affected in *spt6l*, we analyzed the base frequency around the 5' end of RIP-seq reads and previous published Pol V GRO-seq reads[32]. Interestingly, we were able to identify the U feature with the previous GRO-seq but not our RIP-seq (Supplementary Fig. 7a, b). Considering the different strategies in library preparation may dilute the ratio of sliced Pol V transcripts, we divided all Pol V peaks into four groups based on the ratio of U-10 reads in each peak with GRO-seq data (Fig. 6e and Supplementary Data 5) and compared the ratio of U-10 reads in each group with our RIP-seq data. As shown in Fig. 6e, a weak

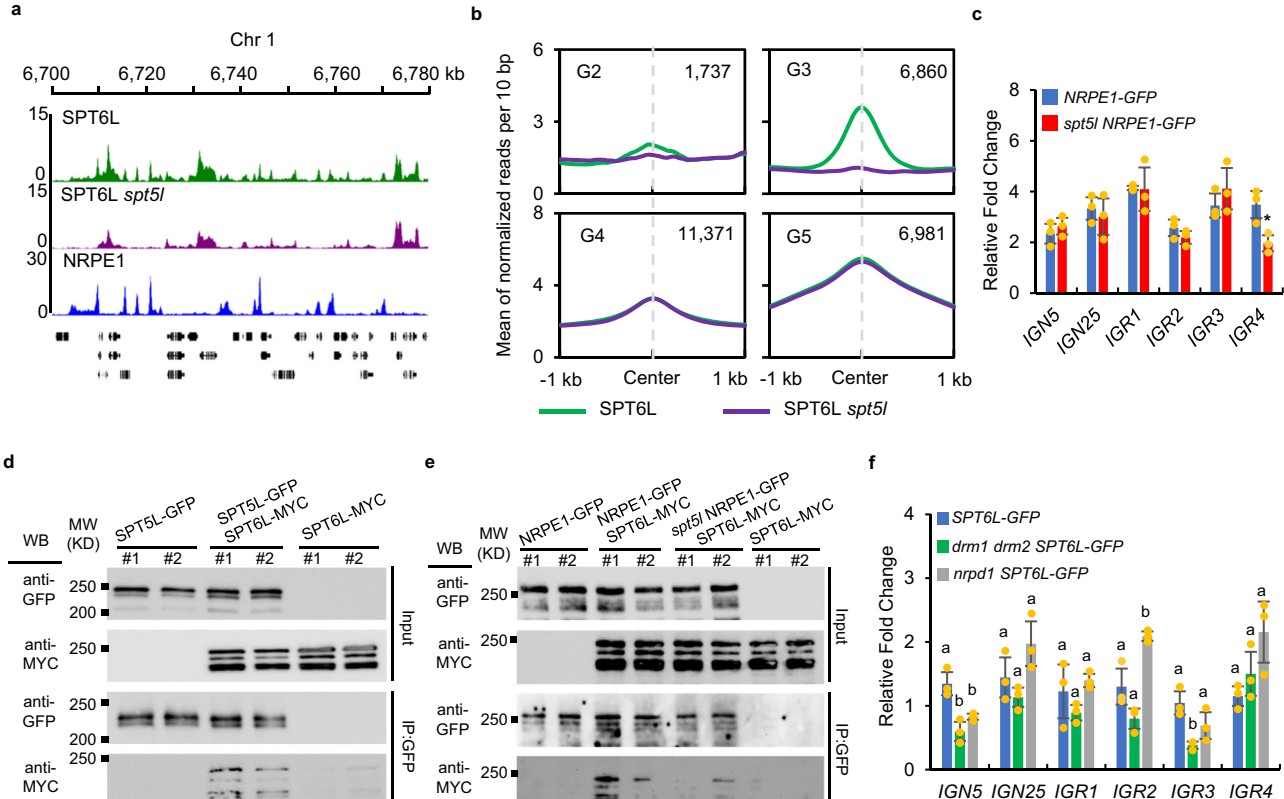

**Fig. 5 | SPT5L is required for the intergenic enrichment of SPT6L. a** Genome tracks display the ChIP-seq signals of SPT6L, SPT6L *spt5l*, and NRPE1 on chromosome 1 (Chr1: 6700–6780 kb). Each sample was averaged over two biological replicates. **b** Binding profiles of SPT6L in *WT* and *spt5l* at four previously defined genomic groups. The ChIP signal of each sample was averaged over two biological replicates. The regions were plotted as indicated in Fig. 1c. **c** ChIP-qPCR showing genomic occupancy by NRPE1-GFP fusion protein in *NRPE1-GFP* and *spt5l NRPE1-GFP*. All the fold changes are relative to ChIP signal obtained at *ACT7* in each sample and replicates. Error bars are presented as mean values ± s.d. from three biological replicates. All significant differences were indicated with *$P < 0.05$, **$P < 0.01$ (unpaired, two-tailed Student's *t*-test). **d** Co-IP examined the interaction between SPT6L and SPT5L. Immunoprecipitation (IP) and Western blot (WB) were performed using specified antibodies (IP: anti-GFP, KTSM1301; WB: anti-Myc, ab9106, anti-GFP, Yeasen, 31002ES60). Data from two biological replicates were shown. **e** Co-IP examined the role of SPT5L in the interaction between SPT6L and NRPE1. IP and WB were performed using specified antibodies (same as Fig.5d). Data from two biological replicates were shown. **f** ChIP-qPCR showing genomic occupancy by SPT6L-GFP fusion protein in *SPT6L-GFP*, *drm1 drm2 SPT6L-GFP*, and *nrpd1 SPT6L-GFP*. All the fold changes are relative to ChIP signal obtained at *ACT7* in each sample and replicates. Error bars are presented as mean values ± s.d. from three biological replicates. Different lowercase letters indicate significant differences, as determined by one-way *ANOVA*, $P < 0.05$.

but similar reduction of U-10 reads ratio in four groups were observed with both *NRPE1-GFP* and *spt6l NRPE1-GFP* RIP-seq reads but not random reads, indicating that the U-10 feature can be weakly revealed in our RIP-seq and the mutation of SPT6L may not affect the slicing process. Altogether, these results indicate that SPT6L is required for sustaining and promoting the transcription of Pol V.

## Discussion

The two plant-specific RNA polymerases Pol IV and V play essential roles in the RdDM pathway. Many accessories of these two polymerases were successfully identified in the last two decades, but it is still not clear how the transcription process of Pol IV and V are regulated in vivo. In this work, we reported the physical association of a conserved Pol II elongator, SPT6L, with the Pol V complex and investigated the roles of SPT6L in the regulation of DNA methylation and Pol V transcription. Our findings indicate a conserved transcription regulation mechanism between these two transcription complexes. Although this is not totally surprising as several Pol II and V shared factors such as AGO4[27], RDM1[37], and RDM4[38] have been identified, it is rather exciting in that SPT6L is the first elongation factor found to play such an important role.

Pol V-dependent DNA methylation serves as the main mechanism to repress the transcription of both TEs and downstream genes[12]. Indeed, knocking out NRPE1 resulted in much-marked up-regulation of Pol V-proximal genes[39]. Interestingly, the co-occupancies of SPT6L and NRPE1 were mainly detected at TSS-distal (−600 to −200 bp upstream of TSS) rather than TSS-proximal (−200 bp to TSS) regions (Fig. 1d). By knocking out NRPE1 and blocking the SPT6L-Pol II interaction, we found an increased enrichment of SPT6L at the nearest downstream TSS of NRPE1 (Fig. 2d) and NRPE1 binding sites (Fig. 2e), respectively. These results suggest that Pol II may directly compete with Pol V in the recruitment of SPT6L and then lead to the low enrichment of SPT6L at TSS-proximal NRPE1 loci. Future works are needed to test the potentially mutual regulation between Pol II and V in the competition for core transcription accessories.

Loss of Pol V mainly causes the reduction of DNA methylation at CHG and CHH[39]. While the requirement for Pol V on the intergenic enrichment of SPT6L (Fig. 2c), the reductions of DNA methylation in *spt6l* were detected in all three contexts (CG, CHG, and CHH) (Fig. 3b, d). According to the amounts of siRNAs, the reduction of DNA methylation in *spt6l* is unlikely resulted from the alteration of siRNA production. The general reduction of mCHG and mCHH is partially contributed by the down-regulation of *CMT2* (Fig. 3e) and mis-regulated Pol V transcripts (Fig. 6a–d). Referring to the decreased mCG, we found the reduced mCG was mainly detected in the NRPE1-bound regions (Fig. 3d) and the decreased mCG in *spt6l* was partially recovered in *spt6l nrpe1* (Fig. 3b, d), suggesting a negative effect of NRPE1 in *spt6l* on the level of mCG. As the binding profile of NRPE1 was

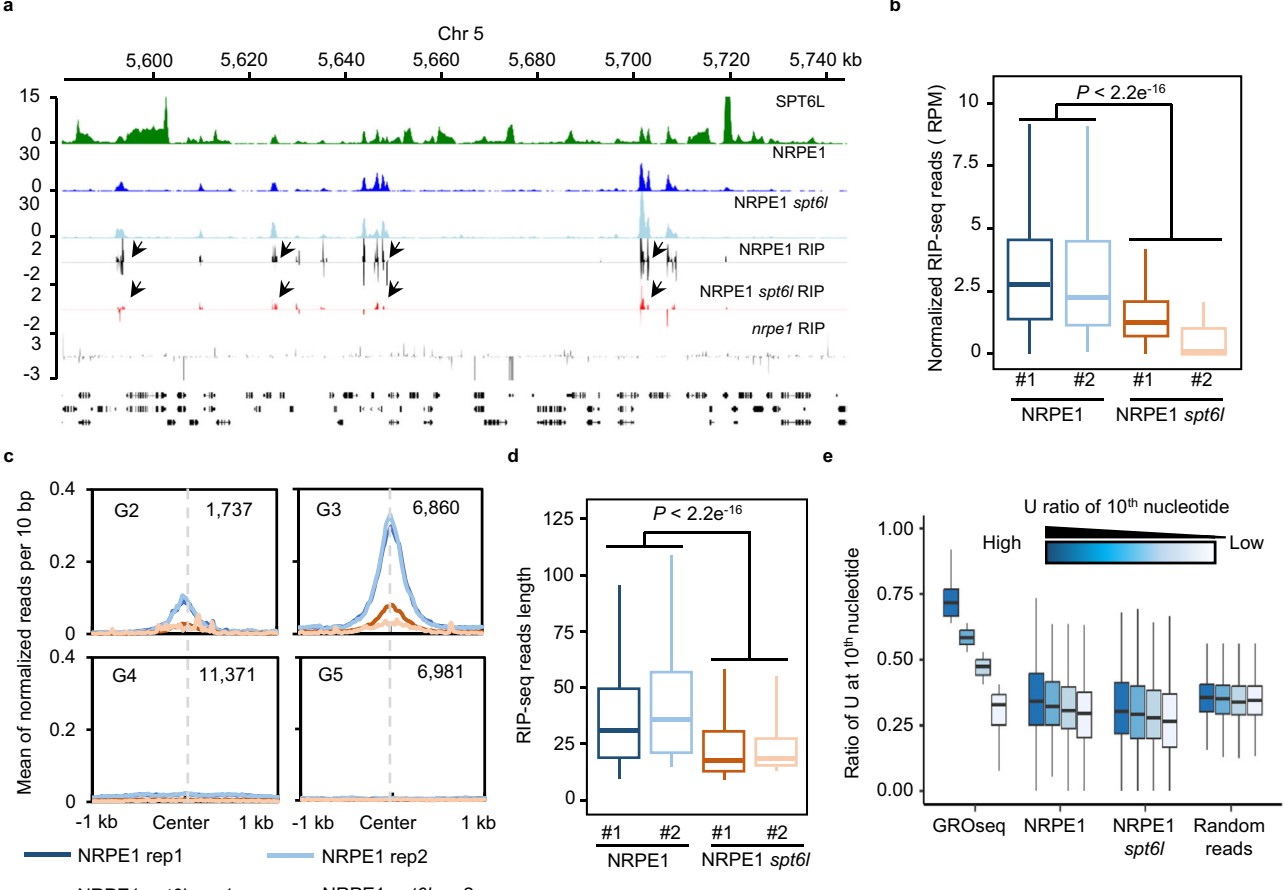

**Fig. 6 | SPT6L is required to maintain Pol V transcripts. a** Genome tracks display the ChIP-seq and RIP-seq signals on chromosome 5. The RIP-seq signals of each sample from different strands were displayed as positive and negative values. The signals from *nrpe1* RIP-seq sample were serves as background. The black arrows pointed to the loci with changed NRPE1 RIP signals in *spt6l*. The ChIP-seq or RIP-seq signals of each sample were averaged over two biological replicates. **b** Boxplots showed the amount of Pol V transcripts in *NRPE1-GFP* and *spt6l NRPE1-GFP* within Pol V peaks. The reads number were normalized to total mapped RIP-seq reads of each sample. Unpaired Wilcoxon test was performed. $P < 2.2e^{-16}$. Two biological replicates were included. **c** Profiles of normalized NRPE1 RIP-seq reads from *NRPE1-GFP* and *spt6l NRPE1-GFP* within four previous defined genomic groups (Fig. 1c). Two biological replicates were included. **d** Boxplots showed the reads length of NRPE1 RIP-seq from *NRPE1-GFP* and *spt6l NRPE1-GFP*. Unpaired Wilcoxon test was performed. $P < 2.2e^{-16}$. Two biological replicates were included. **e** The Ratio of reads containing uridine (U) at position 10 (U-10 reads) in different Pol V peak groups. The Pol V peaks were divided into four groups base on the ratio of U-10 GRO-seq reads in each Pol V peaks (see Bioinformatic analysis). The ratio of U-10 reads generated from GRO-seq (SRR5681049 and SRR5681053), NRPE1 RIP, *spt6l* NRPE1 RIP, and Random reads were plotted. The plotted values are averaged over two biological replicates (Random sample was averaged over three random shuffles). The center line of boxplot, median; box limits, upper and lower quartiles; whiskers, 1.5× interquartile range.

unaffected in *spt6l*, the occupancy of NRPE1 may block the access of CG methyltransferases such as MET1. In the future, it will be interesting to examine how the different types of DNA methylation are mutually affected to each other.

The Pol IV and V shared subunit, NRP(D/E)4, physically interacts with SPT6L (Supplementary Fig. 2f), suggesting that SPT6L may potentially mediate Pol IV functions. However, the mutation of SPT6L has little effect on the proportion and amount of 24-nt siRNA (Fig. 3f, g), which is mainly generated by Pol IV, indicating that. SPT6L is not involved in Pol IV-mediated siRNA biogenesis. Furthermore, distinct from dramatic decrease of SPT6L occupancy in *nrpe1* and *spt5l*, only two of the six loci showed reduced SPT6L occupancy in *nrpd1* (Fig. 5f), indicating that Pol IV is not involved in the intergenic recruitment of SPT6L.

The SPT6L was computationally characterized as one of the top 3 proteins that contained WG/GW repeats[30], a well-known domain to interact with AGOs[30]. However, the SPT6L-WG/GW contributed to neither the intergenic enrichment of SPT6L nor the DNA methylation at RdDM loci (Fig. 4b, f), suggesting that this repeat may be dispensable in DNA methylation. The simultaneous deletion of WG/GW repeats both in NRPE1 and SPT5L reduces the level of AGO4 enrichment and DNA methylation to that of in *nrpe1-11*[29], suggesting that the presence of SPT6L-WG/GW has little contribution to the recruitment of AGO4 and the DNA methylation at RdDM loci.

In the RdDM pathway, Pol V and its transcripts provide a docking platform for downstream components[40]. The mutation of SPT5L dramatically reduced the intergenic enrichment of SPT6L (Fig. 5b) and compromised the association between SPT6L and NRPE1 (Fig. 5e), suggesting that Pol V downstream events rather than Pol V itself determines the intergenic recruitment of SPT6L. The Pol V complex with SPT5L being recruited to it may represent an active state of Pol V, which can further recruit other accessory components such as SPT6L. SPT5L is a homolog of SPT5, which physically contacts SPT6 through its KOW domain in animal cells[22]. In line with this association, a physical interaction between SPT6L and SPT5L was also detected in its N-terminal, which contains the KOW domain (Supplementary Fig. 6c). Interestingly, SPT4, another interacting partner of SPT5, is also involved in the regulation of DNA methylation[41], suggesting that SPT6L may not be the only Pol II and V shared elongators.

Our NRPE1 RIP-seq identified the weak enrichment of Pol V transcript 5′ ends at purines (A/G) (Supplementary Fig. 7b), but the potential bias of template switching in library preparation may also contribute to this feature[42]. Thus, cautions need to be taken when drew conclusion about the 5′-end feature of Pol V transcripts. In addition, we did not detect the preference at +10 within Pol V transcripts, which was revealed previously in another study through GRO-seq[32]. This inconsistency likely results from the different strategies used in library preparation. In the previous GRO-seq, 5′ monophosphorylated (5′-p) RNAs were selectively enriched for library preparation[32], while we generated the Pol V RIP library by template switching, which was able to rescue multiple 5′end of RNA such as 7-methylguanosine capped, 5′ phosphates, and 5′ hydroxyl RNAs[42]. Based on the RNA levels of several *IGN* loci, previous work estimated that about 70% of Pol V transcripts with 5′-triphosphate (5′-ppp) end and 30% transcirpts with 5′-p[43]. Thus, the inconsistent feature may represent different states of Pol V transcripts.

The role of SPT5L in Pol V transcription is still in debate. Previous RT-PCR[44] and IPARE[31] data showed unchanged Pol V transcripts in *spt5l*. The GRO-seq results[32], on the other hand, revealed the roles of SPT5L both in slicing and the amount of Pol V transcripts. In this work, we show that SPT5L plays an essential role in determining the intergenic association of SPT6L (Fig. 5b), and the mutation of SPT6L reduced the amount and length of Pol V RIP reads (Fig.6a–d), suggesting that SPT5L may be involved in the regulation of Pol V transcripts. The inconsistency may due to our modifications to the original IPARE protocol, in which the sonication step was replaced by DNase I treatment. Future works may need to clarify the role of SPT5L in Pol V transcription.

## Methods
### Plant materials and growth
All *Arabidopsis* seeds were stratified for 2 d at 4 °C in darkness. The seeds were then sown on soil or agar plates containing 2.22 g/L Murashige and Skoog (MS) nutrient mix (PhytoTech LABS, M519), 1.5% sucrose (pH 5.8), and 0.8% agar. Plants were grown in growth rooms with 16-h light/8-h dark cycles at 22 °C. All Arabidopsis lines used in this study were in Columbia (Col-0) background. The mutants of *nrpd1-3* (SALK_083051)[34], *nrpe1-11* (SALK_029919)[45], *drm1/2* (CS16383)[46], and *spt6l* (SALK_016621)[9,10] were previously reported. The seeds of *spt5l-1* (SALK_001254C)[47] and *ProNRPE1:NRPE1-FLAG* (CS66156)[48] were obtained from the Arabidopsis Biological Resource Center (ABRC). The seedlings of *spt6l* and *nrpe1-11 spt6l* homozygous were respectively selected from *spt6l*[+/−] and *nrpe1-11 spt6l*[+/−] progenies based on its defected phenotypes as reported previously[9]. To obtain the *nrpe1 spt6l NRPE1-GFP* seedlings, we transformed the *ProNRPE1:NRPE1-GFP* construct into *nrpe1 spt6l*[+/−] plants and selected the correct seedlings from the progenies of homozygous *nrpe1 spt6l*[+/−] *ProNRPE1:NRPE1-GFP*. The transgenic lines *ProSPT6L:SPT6L-GFP* and *ProSPT6L:SPT6L-MYC* were previously reported[9,11]. The transgenic lines of *SPT6L-MYC NRPE1-GFP*, *SPT6L-MYC NRP(D/E)4-GFP*, and *SPT6L-MYC NRPE7-GFP* were generated by transforming *ProNRPE1:NRPE1-GFP*, *ProNRP(D/E)4:NRP(D/E)4-GFP*, and *ProNRPE7:NRPE7-GFP* construct into *SPT6L-MYC* line. All materials used in this study were 10 days old otherwise specified elsewhere.

### Plasmid construction
For the generation of transgenic plants, the full-length *NRPE1* and *SPT5L* genomic region and their -2 kb upstream putative promoters were amplified and cloned into the *pMDC107*. Firstly, part1 (from 2209 to 8315 bp, relative to ATG) and part2 (from −2088 to 2220 bp) of *NRPE1*, part1 (from −2124 to 3675 bp) and part2 (from 3655 to 6582 bp) of *SPT5L* were PCR-amplified from genomic DNA. And then, both part1 of *NRPE1* and *SPT5L* were inserted into *pMDC107* individually by using PmeI and AscI. Finally, part2 of both genes were subcloned into *pMDC107-part1* by ClonExpress® II One Step Cloning Kit

(Vazyme, C112). In addition, the genomic sequences contained upstream regulatory sequence of *NRP(D/E)4* (from −2096 to 1249 bp) and *NRPE7* (from −3012 to 534 bp) were amplified and cloned into the *pMDC107*. For yeast-two-hybrid assay, the CDS of *NRPE1*, *NRP(D/E)2*, *NRP(D/E)4*, *NRPE5*, and *NRPE7* were amplified and cloned into *pGADT7*. Truncated fragment of *SPT5L* CDS were amplified and cloned into *pGADT7* according to previous works[44,49]. The CDS sequence of *SPT6L* was amplified and cloned into *pGBKT7*. All primers used for plasmid construction are listed in Supplementary Table 1.

### Y2H analysis
The vector for bait (*pGBKT7*) and prey (*pGADT7*) were co-transformed into yeast strain AH109 that was selected on medium lacking leucine (Leu) and tryptophan (Trp). Positive colonies were picked up and dropped on -Leu/-Trp/-His medium containing 10 mM E-amino-1, 2, 4 triazol (3-AT) for image recording.

### Confocal microscopy
To detect green fluorescence signals, root tips were cut from 7-day-old seedlings and transferred onto glass slides with 50 μL H$_2$O. The green fluorescence was detected by confocal microscopy (Leica) with excitation at 488 nm and emission at 505–525 nm.

### Immunoblotting and Co-immunoprecipitation
Two hundred milligrams of 10-day-old seedlings were harvested and homogenized to fine powder, which was subsequently dissolved in 300 μL lysis buffer (100 mM Tris−HCl pH 7.5, 300 mM NaCl, 2 mM EDTA, 0.5% TritonX-100, 10% glycerol, 1 mM PMSF, and protease inhibitor cocktail) for 30 min at 4 °C with gentle shaking. Next, the crude lysate was centrifuged at 18,000 g for 10 min at 4 °C to remove debris. For Western blot (WB), the supernatants were mixed with 4× SDS loading buffer and loaded onto SDS-PAGE gels. For Co-IP, we added 25 μL anti-GFP nanobody agarose beads (KT HEALTH, KTSM1301) to the supernatants and incubated for 2 h at 4 °C with gentle shaking. The beads were washed five times with wash buffer (100 mM Tris−HCl (pH 7.5), 300 mM NaCl, 2 mM EDTA, and 0.75% TritonX-100). After centrifugation, the beads were boiled with 2× SDS sample buffer for 5 min. The interacting proteins then loaded onto SDS-PAGE gels. For WB in yeast, the AH109 cells transformed with AD-NRPEx were grown to OD600 ≈ 0.6 in medium lacking leucine (Leu) and tryptophan (Trp). And, the following procedures were performed as previously described[50]. The antibodies used for WB are listed as follow: anti-GFP (Yeasen, 31002ES60; 1:10,000 dilution), anti-H3 (Abcam, ab1791; 1:20,000 dilution), anti-MYC (Abcam, ab9106; 1:20,000 dilution), anti-HA (Vazyme, RA1004; 1:5,000 dilution).

### Chromatin immunoprecipitation and library preparation
For most of the ChIP samples, ChIP assays were carried out as previously described[11]. For NRPE1-Flag ChIP, the nuclei were firstly enriched as previously described[51] and then followed with nuclei lysis. Immunoprecipitation was performed by using either anti-GFP antibody (Abcam, ab290) or anti-FLAG M2 Magnetic Beads (Sigma−Aldrich, M8823). For ChIP-qPCR, at least two biological replicates were included and primers were listed in Supplementary Table 1. For ChIP-seq, the libraries of ChIP DNA were prepared following the published protocol[52] with at least two biological replicates otherwise specified elsewhere. The reads information of different sample was collected in Supplementary Table 2. The correlations across biological replicates can be found in Supplementary Fig. 8a.

### RNA and small RNA-seq
Total RNA was extracted from 10-day-old seedlings using TRIzol (Invitrogen, 15596-018). Genomic DNA was removed by treating with TURBO DNase and then the DNase inactivated RNA was used for either mRNA or small RNA library preparation. For small RNA, RNA samples

were separated on a PAGE gel, and the 18- to 30-nt fraction of the gel was cut for small RNA purification. For RNA-seq, Poly(A) mRNAs was enriched with NEBNext Poly(A) mRNA Magnetic Isolation Module (NEB). Library preparation and sequencing were performed using Illumina reagents according to the manufacturer's instructions. The correlations across biological replicates can be found in Supplementary Fig. 8b.

### Chop-PCR and whole-genome bisulfite sequencing

Genomic DNA was extracted from 10-day-old seedlings using the DNeasy Plant Mini Kit (QIAGEN 69104). And then, about 100 ng genomic DNA was digested overnight with methylation-sensitive restriction endonucleases (Hae III, NEB, R0108S). The digested DNA was used to amplify the indicated regions by PCR using primers flanking the endonuclease recognition sites. Primers are listed in Supplementary Table 1. For the bisulfite sequencing, the extracted DNA was directly sent to the NovoGene for whole-genome bisulfite sequencing (WGBS). The correlations across biological replicates can be found in Supplementary Fig. 9.

### RNA immunoprecipitation and library preparation

The NRPE1 RIP-seq was performed as previously described[31,53,54] with modifications. Briefly, 2 g of 10-day-old seedlings was used for chromatin extraction. Chromatin was treated by DNase I (NEB, M0303S) for 1 h, and then we added 1% final concentration SDS to the treated Chromatin. Supernatant was diluted five times with chip diluent buffer. And IP was performed using 2 μL/IP anti-GFP (ab290, Abcam) at 4 °C for overnight. After rescuing Pol V-associated RNA, the removal of residual gDNA and addition of poly-A tail were performed as described in IPARE protocol[31]. To increase the efficiency of reverse transcription 10 times higher amount of dCTP (0.5 μl 100 mM) was added into reverse transcription buffer[42]. The following DNA purification and library preparation were similar to that in IPARE protocol[31].

### Bioinformatic analysis

**ChIP-seq.** The adaptors of raw ChIP-seq reads were removed by using cutadapt[55] (version 3.4, default settings) and mapped to the Arabidopsis genome by Bowtie2[56] (version 2.4.2, default settings) in pair-end mode. The unmapped, improperly paired, and duplicated reads were removed using samtools[57] (version 1.11, default settings). And then, bam files were converted to BEDPE format with bedtools (version 2.27.1, default settings), and the pair-end mode of MACS2[58] (2.2.7.1, -f BEDPE, -g 135000000, -q 0.001) was used to generate peak lists. The high confident peak list across two biological replicates was generated by using IDR (version 2.0.3, --idr-threshold 0.01). The bamCoverage of deeptools[59] (version 3.5.1, -bs 10, --normalizeUsing RPGC, --effectiveGenomeSize 135000000) was used to generate genome coverage files. The values under heatmaps and plots were generated with computeMatrix (subcommand of deeptools, -bs 10 –missingDataAsZero). The analyses of correlations between samples/replicates were performed by using multiBigwigSummary (subcommand of deeptools, BED-files mode, the regions of correlation analyses for NRPE1 and SPT6L IP samples were NRPE1 and SPT6L peaks, respectively). The different states of the genome were identified by using ChromHMM[60] (version 1.24, default settings). The averaged coverage file from two biological replicates was generated by running a GitHub script (http://wresch.github.io/2014/01/31/mergebigwig-files.html). Genome tracks were generated with pyGenomeTracks[61].

**BS-seq.** The raw reads of BS-seq were processed by cutadapt to remove adaptors and aligned to Arabidopsis genome by using Bismark[62] packages (version 0.23.1, default settings). PCR duplicates were removed by using deduplicate_bismark (default settings). And then, the methylated bases were extracted by using bismark_methylation_extractor (default settings). Finally, the outputs of

bismark_methylation_extractor can be load into a R package-methylkit[63] (version 1.22.0, mincov = 4, win.size = 500, step.size = 500, difference = 25, qvalue = 0.01) to identify differential methylated regions and calculate correlation values.

**RNA-seq.** The cleaned RNA-seq raw reads were mapped into Arabidopsis genome with STAR (version 2.7.11a, default settings, --genomeSAindexNbases 12) and the raw reads count per gene in each sample was calculated by RSEM package (version 1.3.3, with default settings). Finally, the analysis of differential expression was performed by using an R package-DEseq2 (version 3.18).

**smRNA-seq.** The reads quality of smRNA-seq was firstly checked by using FastQC (version 0.11.9) and the adaptor and linker were removed by using cutadapt. And then, the processed reads were mapped into *Arabidopsis* genome with Bowtie2 (default settings). After filtering out the unmapped reads, the mapped reads were converted to bed format with bedtools. Different sizes of small RNAs were selected, counted, and compared within different genome regions.

**RIP-seq.** The trimming, mapping, and removing PCR duplicates were performed as previously described[31] with minor modifications. Briefly, the 8 bp long unique molecular identifier (UMI) in the first read of paired-end reads were removed and appended to the read name by using UMI-tools[64] (version 1.1.2). And then, the reads of read 1 (first read in paried-end reads) were then trimmed to remove the 3'poly (A) and 5' TATAGGG (cutadapt, -m 10). Finally, the processed reads (read 1) were mapped Arabidopsis genome with default settings. The PCR duplicates were removed by using the UMI-tools and processed reads were converted to bed format with bedtools. To identify Pol V transcripts, the mapped RIP reads from NRPE1 or *spt6l* NRPE1 were intersected with reads identified in *nrpe1* and only the non-overlapped reads were kept. Finally, the processed RIP-seq reads were overlapped with NRPE1 peaks, and only the reads within NRPE1 peaks were considered as Pol V transcripts. To calculate the ratio of U-10 reads in each Pol V peak, GRO-seq reads (SRR5681049 and SRR5681053) were mapped and filtered as previously described[32]. The Final Pol V-dependent GRO-seq reads were overlapped with Pol V peaks (identified in this work). To avoid the low reads bias on the U-10 reads ratio, the Pol V peaks with less than 10 GRO-seq and NRPE1 RIP reads were removed. To assess the potential sequence bias in grouped Pol V peaks, the NRPE1 RIP reads within each Pol V peak were randomly shuffled within the peak (3 times). The final grouped Pol V peaks with U-10 reads number and ratio can be found in Supplementary Dataset 5.

### Reporting summary

Further information on research design is available in the Nature Portfolio Reporting Summary linked to this article.

## Data availability

The data that support the findings of this study are available from the corresponding author upon request. The high-throughput sequencing data generated in this study have been deposited in Gene Expression Omnibus with the accession code GSE233781. Source data are provided with this paper.

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

## Acknowledgements

We thank the Arabidopsis Biological Resource Centre for providing the mutant seeds used in this study. The mutant seeds of *nrpd1-3*, *nrpe1-11*, and *drm1/2* were kindly provided by Dr. Shulin Deng. This work was supported by the National Natural Science Foundation of China to C.C. (32070648) and J.S. (32100474), Guangdong Pearl River Talent Program to C.C. (2021QN020018), Science and Technology Projects in Guangzhou to C.C. (E3330900-01), and the Natural Science and Engineering Council of Canada to Y.C. (RGPIN/04625-2017).

## Author contributions

C.C. conceived the project; C.C., C.H.W., and Y.J.L. designed the experiments; N.D. performed SPT6LdeltaWG/GW-related ChIP-seq and immunoblots; J.S. performed the all the ChIP-qPCR assays; Z.Z. examined qPCR and Chop-PCR experiments; Y.J.L. performed all the rest of the experiments; Y.J.L., J.Y.L., and J.L. analyzed ChIP-seq data; C.C. and Y.H.C. analyzed rest of high-throughput data; C.C. and Y.J.L. wrote the paper.

## Competing interests

The authors declare no competing interests.
