## [Peer Review File · Nature Communications]

A conserved Pol II elongator SPT6L mediates Pol V transcription to regulate RNA-directed DNA methylation in ArabidopsisREVIEWER COMMENTS

Reviewer #1 (Remarks to the Author):

The RNA-directed DNA methylation is a plant specific pathway that requires the transcripts from Pol IV and Pol V. Despite the extensive studies on these two Pols, the regulation of them remains less studied. In this paper, Liu et al. identified a Pol II elongation factor SPT6L that also coordinates the function of Pol V, suggesting a new RdDM regulation factor. The authors first noticed the SPT6L by the unusual SPT6L ChIP-seq signal from previous data which raise the potential crosstalk between SPT6L and Pol V. The direct interaction between SPT6L and Pol V was confirmed and the genetic correlation was later confirmed, too. By extensive analysis, the author proposed that the SPT6L is responsible for the Pol V elongation. Overall, it is an interesting story that identified novel component of RdDM to regulate Pol V.

The major concern is about the interaction part. The Y2H indicate SPT6L interacts with NRPE3B, NRPE4, and NRPE7 but not NRPE1, while the co-IP identified all the four proteins. As the Pol V is stable complex, if SPT6L interacts with any single subunit of Pol V, it may IP the intact Pol V, including all the 12 subunits. More over, the single subunits of Pol V may not stable enough to interact with SPT6L, so the negative results also need attention. I suggest the author to focus on the overall interaction between SPT6L and Pol V, and to avoid misleading by the Y2H and co-IP data.

Reviewer #2 (Remarks to the Author):

In this manuscript, the authors claim that SPT6L regulates RNA-directed DNA methylation in *Arabidopsis thaliana* by mediating Pol V transcription elongation, based mainly on (1) re-analysis of previous ChIP data on SPT6L and Pol V as well as new datasets using newly created transgenic lines, (2) probing interaction of SPT6L and Pol V complex by co-immunoprecipitation and yeast two hybrid approaches and (3) genome-wide analysis of Pol V-associated RNA in wildtype and *spt6l* by RNA-immunoprecipitation. Although the idea of a transcription elongation factor shared between Pol II and Pol V would be intriguing if true, there are numerous concerns that raise doubts about the claims, as outlined below.

Major concerns

- *spt6l* mutant causes pleiotropic defects. And as the authors stated in the manuscript, the levels of 4 out of 5 DNA methyltransferases in *Arabidopsis* are reduced in the *spt6l* mutant. So, it is possible that the hypomethylation in *spt6l* is due to reduced accumulation of methyltransferases instead of mis-regulation of Pol V. The observation that the genome-wide methylation pattern of *spt6l* is different from the patterns in each of DNA methyltransferase single mutants is not sufficient to rule out this concern.
- The genomic data is difficult to interpret. Even though the authors presented Pol V, Pol II, and SPT6L ChIP-seq results in the manuscript, it is unclear how many SPT6L peaks overlap with Pol II and Pol IV. Dissecting the genome into 6 groups (S1 to S6 in the manuscript) does not help the reader to understand the interaction sites of SPT6L. A more defined classification of genomic regions, such as genic regions, intergenic regions, and TEs, etc., would be more useful. "TSS" in this paper's context is also confusing, as it represents Pol II's TSS sites, not Pol V's.
- The authors used yeast two-hybrid (supplementary figure 2F/6C) assays to probe interactions between SPT6L and Pol V-specific subunits/SPT5L, but the results overall are not clear, as NRPE4, 7 and SPT5L signals are inconsistent among replicates. NRPE3B (presumably AT2G15400? -this should be clarified) was detected in Pol II by mass spec, as well (Ream TS. et al. 2015 [pmid 25813043]). The Y2H experiments lack proper controls needed to conclude a negative result (ie. Is the NRPE1 AD construct working properly?).
- The rationale for probing Pol V specific subunits, but not Pol II-specific subunits, or shared subunits, is not clear. About half of the subunits of Pol II and Pol V are shared, and SPT6L was shown to associate with Pol II in previous studies by the authors' group. The RPB1-CTD is one

characterized interaction interface, but cryoEM structures of mammalian Pol II complexes containing SPT6 (for example, see pdb: 6gmh) show that SPT6 binds the RPB4/7 stalk in addition to the RPB1-CTD. Is this also true for Arabidopsis Pol II – SPT6L? More subunits should be tested.

- To support the claim that SPT6L is involved in Pol V transcription elongation, measurements of full-length Pol V transcripts would be needed. Accumulating evidence indicates that Pol V transcripts are sliced by AGO4. For the RIP-Seq experiment, it is unclear if the Pol V-bound RNAs are sliced or not. To study the size of full-length Pol V transcripts, the authors should do the RIP experiments in ago4 or ago4/6/9 mutant backgrounds.
 - o the sequence logo values in figure 6E are too small to make meaningful conclusions.
 - o Figure 6A should include a ChIP track of NRPE1 spt6l for this particular region, to validate comparable NRPE1 loading for the RIP comparison.

Specific comments

I.61: “The short products from Pol IV and V are likely due to the lack of surfaces to recruit Pol II transcription factors such as TFIIB and TFIIS”

This statement is misleading and at odds with what is known. Evidence indicates that Pol IV transcripts are short because of Pol IV’s inability to elongate through double-stranded DNA. Pol V transcript length has only been estimated, not demonstrated empirically. But RT-PCR and other analyses by Wierzbicki and colleagues have suggested that Pol V transcripts may be 200 nt or longer.

I.156-158: The citation to the figures S4 and S5 does not match the context described here. For the Co-IP assay, did the authors cross the NRPE1-GFP and SPT6L-Myc lines? Please clarify in the methods.

Since there are pleiotropic effects in the spt6l mutant, how did the authors generate the gNRPE1-GFP/spt6l line and perform ChIP-Seq? Please clarify in the methods.

Would AGO4 occupancy correlate with SPT6L occupancy in intergenic spaces in the SPT6LΔWG/GW line? Would such data be available?

I.111: FigS2C. – Why is there such a big discrepancy in peak NRPE1 occupancy sites (~7.8k versus ~11.9k) between the current manuscript and the study of Gallego-Bartolome, J. et al. 2019 Cell?

I.151 Fig.S3B – Is WT supposed to be referring to NRPE1-GFP? The WT in the supplementary figure has no NRPE1 signal at all.

Typos

I.65: above structure data -> above structural data

I.93/144: promoted us to -> prompted us to

I.94: the silence of -> the silencing of

I.99: NREP1 -> NRPE1

I.126: NPPE1-SPT6L

I.268/314/386: splicing feature -> slicing feature

Reviewer #1 (Remarks to the Author):

The RNA-directed DNA methylation is a plant specific pathway that requires the transcripts from Pol IV and Pol V. Despite the extensive studies on these two Pols, the regulation of them remains less studied. In this paper, Liu et al. identified a Pol II elongation factor SPT6L that also coordinates the function of Pol V, suggesting a new RdDM regulation factor. The authors first noticed the SPT6L by the unusual SPT6L ChIP-seq signal from previous data which raise the potential crosstalk between SPT6L and Pol V. The direct interaction between SPT6L and Pol V was confirmed and the genetic correlation was later confirmed, too. By extensive analysis, the author proposed that the SPT6L is responsible for the Pol V elongation. Overall, it is an interesting story that identified novel component of RdDM to regulate Pol V.

The major concern is about the interaction part. The Y2H indicate SPT6L interacts with NRPE3B, NRPE4, and NRPE7 but not NRPE1, while the co-IP identified all the four proteins. As the Pol V is stable complex, if SPT6L interacts with any single subunit of Pol V, it may IP the intact Pol V, including all the 12 subunits. More over, the single subunits of Pol V may not stable enough to to interact with SPT6L, so the negative results also need attention. I suggest the author to focus on the overall interaction between SPT6L and Pol V, and to avoid misleading by the Y2H and co-IP data.

Thank you for your comments and suggestion. We agree your point that it is wired to emphasize the difference between Y2H and Co-IP. In this revision, we have reworded the sentences and pay more attention to complex interaction rather than specific subunit. In addition, we have repeat the Y2H three more times and only the interaction between SPT6L and NRPE4 can be stably detected. The interaction between NRPE7 and SPT6L is very weak and unstable, thus, we will not claim the interaction between them. Referring to NRPE3B, we thank to reviewer2's comment that this protein has been detected in Pol II complex. Thus, we do not mention it in this revised manuscript.

Reviewer #2 (Remarks to the Author):

In this manuscript, the authors claim that SPT6L regulates RNA-directed DNA methylation in *Arabidopsis thaliana* by mediating Pol V transcription elongation, based mainly on (1) re-analysis of previous ChIP data on SPT6L and Pol V as well as new datasets using newly created transgenic lines, (2) probing interaction of SPT6L and Pol V complex by co-immunoprecipitation and yeast two hybrid approaches and (3) genome-wide analysis of Pol V-associated RNA in wildtype and spt6L by RNA-immunoprecipitation. Although the idea of a transcription elongation factor shared between Pol II and Pol V would be intriguing if true, there are numerous concerns that raise doubts about the claims, as outlined below.

Major concerns

- *spt6l* mutant causes pleiotropic defects. And as the authors stated in the manuscript, the levels of 4 out of 5 DNA methyltransferases in Arabidopsis are reduced in the *spt6l* mutant. So, it is possible that the hypomethylation in *spt6l* is due to reduced accumulation of methyltransferases instead of mis-regulation of Pol V. The observation that the genome-wide methylation pattern of *spt6l* is different from the patterns in each of DNA methyltransferase single mutants is not sufficient to rule out this concern.

Thank you for your comments. We agree with reviewer that the pleiotropic defects of *spt6l* is not easy to establish the link between changed DNA methylation to one specific effect. Thus, we tried our best to evaluate the potential changes in different parts of DNA methylation pathway, such as siRNA production and methyltransferases expression. In the revision, we performed RNA-seq and comprehensively compared the changes of DNA methylation related genes such as methyltransferases, DNA demethylases, siRNA production, and several key components of RdDM pathway. In addition, we found that the reduced DNA methylation (mCG) was mainly detected in NRPE1 associated regions (Figure 3B and 3D), indicating that the *spt6l*-caused reduction of methylation has region specificity. If the mis-regulation of DNA methyltransferases or demethylases serves as the main contributor, we should expect a general but not specific reduction of methylation. With the above data and rational, we think the mis-regulation of DNA methyltransferases and demethylases are not the main reason of reduced DNA methylation in *spt6l*. It is of note that we found the expression changes of *DRM1* (*AT5G15380*) and *DRM2* (*AT5G14620*) in RNA-seq (Figure 3E) are inconsistent to that of original qPCR. By checking the primers (original Table 1 Primer), we noticed that the original qPCR results were generated by wrong primers, which targeted to *DRM1* (*AT1G28330*) and *DRM2* (*AT2G33830*). To confirm the RNA-seq results, we redesigned primers targeted to *DRM1* (*AT5G15380*), *DRM2* (*AT5G14620*) and found similar changes to that in RNA-seq data.

- The genomic data is difficult to interpret. Even though the authors presented Pol V, Pol II, and SPT6L ChIP-seq results in the manuscript, it is unclear how many SPT6L peaks overlap with Pol II and Pol IV. Dissecting the genome into 6 groups (S1 to S6 in the manuscript) does not help the reader to understand the interaction sites of SPT6L. A more defined classification of genomic regions, such as genic regions, intergenic regions, and TEs, etc., would be more useful. “TSS” in this paper’s context is also confusing, as it represents Pol II’s TSS sites, not Pol V’s.

Thank you for your comments. Originally, we have provided the peak lists in Supplementary Dataset1, which includes SPT6L genic peak (associated with Pol II), SPT6L intergenic peaks (associated with Pol V), NRPE1 and SPT6L overlapped peaks, NRPE1-only peaks. For the overlapping with different genomic feature, we have provided the proportions of TE, gene, and others in Supplementary Figure 1B. To the divided genomic states (6 states), we have provided the enrichment of each state on different genomic features in Supplementary Figure 2E. Plus, in this revision, we further analyzed the categories of TE associated by NRPE1 and NRPE1-SPT6L peaks (Supplementary Figure 1E). For the word “TSS”, indeed, we referred to Pol II transcription starting sites rather than Pol V. The TSS was mainly mentioned in Figure 2D. In this part, we were analyzed the change of Pol II occupancy at the nearest downstream genes of Pol V peak. It was reported that Pol V can affect the expression of downstream genes and the affection was getting obvious with the closing distance between Pol V peak and downstream TSS (Zhong, X. et al, 2012). To make this point clear, we have rephrased the words and sentences.

- The authors used yeast two-hybrid (supplementary figure 2F/6C) assays to probe interactions between SPT6L and Pol V-specific subunits/SPT5L, but the results overall are not clear, as NRPE4, 7 and SPT5L signals are inconsistent among replicates. NRPE3B (presumably AT2G15400? -this should be clarified) was detected in Pol II by mass spec, as well (Ream TS. et al. 2015 [pmid 25813043]). The Y2H experiments lack proper controls needed to conclude a negative result (ie. Is the NRPE1 AD construct working properly?).

Thank you for your comments. We are sorry for missing the important reference you mentioned and we have removed the NRPE3B related results from the revised manuscript. For the Y2H results, we have repeated the experiments with serial dilution and provided corresponding controls in Supplementary figure 2F and 6C.

- The rationale for probing Pol V specific subunits, but not Pol II-specific subunits, or shared subunits, is not clear. About half of the subunits of Pol II and Pol V are shared, and SPT6L was shown to associate with Pol II in previous studies by the authors’ group. The RPB1-CTD is one characterized interaction interface, but cryoEM structures of mammalian Pol II complexes containing SPT6 (for example, see pdb: 6gmh) show that SPT6 binds the RPB4/7 stalk in addition to the RPB1-CTD. Is this also true for Arabidopsis Pol II – SPT6L? More subunits should be tested.

Thank you for your comments. The co-occurrence of SPT6L and Pol V on genome prompted us to examine whether SPT6L can associate with Pol V complex. As Pol V and II have many shared subunits, it is still not clear whether SPT6L can form protein complex with Pol V or not, even we detected the interactions between SPT6L and Pol II-specific or Pol II-V shared subunits. It is possible that the association of SPT6L to Pol V is also contributed by Pol II-V shared subunits, but we think the examination of Pol V-specific subunits will be much more straightforward. It is of interesting to test the interaction between SPT6L and other Pol II subunits in *Arabidopsis*, but in this work, we mainly focus on Pol V-SPT6L rather than Pol II.

- To support the claim that SPT6L is involved in Pol V transcription elongation, measurements of full-length Pol V transcripts would be needed. Accumulating evidence indicates that Pol V transcripts are sliced by AGO4. For the RIP-Seq experiment, it is unclear if the Pol V-bound RNAs are sliced or not. To study the size of full-length Pol V transcripts, the authors should do the RIP experiments in *ago4* or *ago4/6/9* mutant backgrounds.

Thank you for your comments. To our knowledge, it is still not clear whether AGO4/6/9 can slice Pol V transcript or not. The result from Wierzbicki's lab (Masayuki Tsuzuki, 2020) indicates that the Pol V transcript is general unchanged in *ago4* mutant. Their following up work (M. Hafiz Rothi, 2021) further divided the Pol V transcripts into AGO4 dependent (I) and independent (II) and concluded that the different levels of DNA methylation between (I) and (II) linked to Pol V transcripts. This result suggests that the slicing role of AGO4 to Pol V transcript, if it exists, will not function equally to all the Pol V transcripts. In our case, the reduced Pol V transcripts has been found within most of Pol V peaks (Figure 6C).

The most direct evidence for the slicing role of AGO4 on Pol V transcripts, to our knowledge, is the reduced GRO-seq signals in *ago4* by comparing the GRO-seq signals from *ago4 nrpel*, *nrpel*, and WT (Liu, 2018). As we mentioned in discussion part, this work prepared their GRO-seq library by exclusively selecting 5' monophosphorylated RNA, which is about 30% of total Pol V transcripts (estimated by Pikaard's lab, Wendte, 2017). Thus, it seems the slicing role of AGO4 may be only effective on a small proportion of Pol V transcripts.

At last, according to current RdDM model, siRNA is required for AGO4 to associated with Pol V/Pol V transcripts. We found the level of siRNA is comparable between WT and *spt6l* (Figure 3F and 3G). And, in AGO4 side, the newly provided RNA-seq data indicated that the expression of AGO4 in WT is similar to that in *spt6l* (Figure 3E), suggesting that the effect of AGO4-slice, if it exists, will be similar in WT and *spt6l*. Thus, the reduced Pol V transcripts and read length in *spt6l* unlikely result from changed AGO4 slicing ability.

Altogether, we think the examination of the Pol V transcripts in *ago4* or even *ago4/6/9* mutant will be interesting, but it will not affect the result of SPT6L's role in Pol V transcription.

o the sequence logo values in figure 6E are too small to make meaningful conclusions.

Thank you for your comments. We agree with the reviewer that the enrichment of G/A at the assembled 5' end of Pol V transcript is too weak. In addition, the enrichment of G/A may also result from the bias of Template-switch technique (Wulf., 2019). Thus, we did not claim the 5' end feature of assembled Pol V transcripts. In the revision, we still kept the sequence logo to indicate the failure of detection +10 U, suggesting that the species of Pol V transcript identified by ours are likely different to previous work (Liu, 2018)

o Figure 6A should include a ChIP track of NRPE1 *spt6l* for this particular region, to validate comparable NRPE1 loading for the RIP comparison.

Thank you for your comments. We have included the track of *spt6l* NRPE1 ChIP in this revised figure.

Specific comments

l.61: "The short products from Pol IV and V are likely due to the lack of surfaces to recruit Pol II transcription factors such as TFIIB and TFIIS"

This statement is misleading and at odds with what is known. Evidence indicates that Pol IV transcripts are short because of Pol IV's inability to elongate through double-stranded DNA. Pol V transcript length has only been estimated, not demonstrated empirically. But RT-PCR and other analyses by Wierzbicki and colleagues have suggested that Pol V transcripts may be 200 nt or longer.

Thank you for your comments. Indeed, our original description is overstatement. We have rephrased the text and introduced the previous *in vitro* data about the biochemical nature of Pol V. The recent structural data just provide further explanation why Pol IV and V can not transcribe smoothly.

l.156-158: The citation to the figures S4 and S5 does not match the context described here.

Thank you for your comment. Sorry for the confusion. The S4 and S5 (line: 156-158) in original manuscript refers to genome state 4 and 5. To avoid the potential confusion, we renamed the six genome states and labeled the states as G1 to G6 in the revised manuscript.

For the Co-IP assay, did the authors cross the NRPE1-GFP and SPT6L-Myc lines? Please clarify in the methods.

Thank you for your question. The answer is NO. For the Co-IP assays including NRPE1, NRPE4, NRPE7, and SPT5L, we were direct transformed the corresponding constructs into SPT6L-Myc lines. This part already added into the revised manuscript.

Since there are pleiotropic effects in the *spt6l* mutant, how did the authors generate the gNRPE1-GFP/*spt6l* line and perform ChIP-Seq? Please clarify in the methods.

Thank you for your question. Indeed, the *spt6l* mutant can not grow further. Thus, we transformed *gNRPE1-GFP* into *nrpe1 spt6l^{+/+}* plants and obtained *nrpe1 spt6l NRPE1-GFP* seedlings from the progenies of homozygous *nrpe1 spt6l^{+/+} NRPE1-GFP* (homozygous for transgene). We have to germinate many seeds (theoretically, one quarter of them is *spt6l* mutant) to get enough material for ChIP and RIP experiments. This part already added into the revised manuscript.

Would AGO4 occupancy correlate with SPT6L occupancy in intergenic spaces in the SPT6LΔWG/GW line? Would such data be available?

Thank you for your questions. As the binding pattern of SPT6LΔWG/GW similar to that in SPT6L, we plotted the published AGO4 ChIP-seq data (in *fwa* background, Gallego-Bartolome, J. et al. 2019 Cell) along with SPT6L, NRPE1, and Pol II at four chromatin states (shown at below). Although we could observe the enrichment of AGO4 at genic regions, the majority binding signals of AGO4 were detected at S2 and S3 (NRPE1 binding loci), which were similar to that of SPT6L. However, we did not perform the ChIP-seq of AGO4 in *spt6l SPT6LΔWG/GW* background.

We assumed that the reviewer would like to know whether the AGO hook of SPT6L contribute the occupancy of AGO4 or not. In general, we think the occupancy of AGO4 may not affect in *spt6l SPT6LΔWG/GW*. 1, The simultaneous deletion of AGO hook in NRPE1 and SPT5L can abolish the AGO4 enrichment at RdDM loci (Lahmy, S. et al. 2016); 2, The N-terminal rather than C-terminal of SPT5L interacted with SPT6L. So,

SPT6L may still associated with Pol V complex after the AGO hook (NRPE1 and SPT5L) deletion in previous simultaneous deletion, suggesting SPT6L may not be involved in the recruitment of AGO4. Altogether, we feel that the genome-wide profile of AGO4 in *spt6l SPT6LΔWG/GW* may be similar to that in WT.

l.111: FigS2C. – Why is there such a big discrepancy in peak NRPE1 occupancy sites (~7.8k versus ~11.9k) between the current manuscript and the study of Gallego-Bartolome, J. et al. 2019 Cell?

Thank you for your comment. This discrepancy is due to different standards in data processing. The published NRPE1 ChIP-seq data (Gallego-Bartolome, J. et al. 2019 Cell) only include one biological replicate. So, the number of peak (~11.9k) was a direct output of MACS2 peak calling software (parameters: -p 0.001). The peak number (~7.8k) reported in this manuscript was generated after removing irreproducible peaks across two biological replicates by running *idr* software with threshold IDR < 0.01. In fact, after peak calling using MACS2 (same parameters), our two biological replicates can obtain similar amounts of peaks to the published data (biological 1: 13,325 and biological 2: 10,351). We just focused on those highly confident peaks.

l.151 Fig.S3B – Is WT supposed to be referring to NRPE1-GFP? The WT in the supplementary figure has no NRPE1 signal at all.

Thank you for your comment. The labeling in Fig.S3B is correct and the WT in this blotting serves as a negative control, which designed to control the real blotting signals rather than non-specific bands.

Typos

l.65: above structure data -> above structural data

l.93/144: promoted us to -> prompted us to

l.94: the silence of -> the silencing of

l.99: NREP1 -> NRPE1

l.126: NPPE1-SPT6L

l.268/314/386: splicing feature -> slicing feature

Thank you for pointing out the above typos, we have made changes to all the related places.

REVIEWER COMMENTS

Reviewer #1 (Remarks to the Author):

My concerns have been addressed.

The supplementary tables look to be printed incorrectly. Please correct them.

Reviewer #2 (Remarks to the Author):

Re-review comments

The revised manuscript has addressed some of the reviewers' concerns, but several major issues are still present (see attachment).

Reviewer1:

My concerns have been addressed.

The supplementary tables look to be printed incorrectly. Please correct them.

Thank you for your comment. The format is corrected in this version.

Reviewer 2:

Major concerns

***Our previous review comments:** spt6l mutant causes pleiotropic defects. And as the authors stated in the manuscript, the levels of 4 out of 5 DNA methyltransferases in Arabidopsis are reduced in the spt6l mutant. So, it is possible that the hypomethylation in spt6l is due to reduced accumulation of methyltransferases instead of mis-regulation of Pol V. The observation that the genome-wide methylation pattern of spt6l is different from the patterns in each of DNA methyltransferase single mutants is not sufficient to rule out this concern.*

Authors' response:

Thank you for your comments. We agree with reviewer that the pleiotropic defects of spt6l is not easy to establish the link between changed DNA methylation to one specific effect. Thus, we tried our best to evaluate the potential changes in different parts of DNA methylation pathway, such as siRNA production and methyltransferases expression. In the revision, we performed RNA-seq and comprehensively compared the changes of DNA methylation related genes such as methyltransferases, DNA demethylases, siRNA production, and several key components of RdDM pathway. In addition, we found that the reduced DNA methylation (mCG) was mainly detected in NRPE1 associated regions (Figure 3B and 3D), indicating that the spt6l-caused reduction of methylation has region specificity. If the mis-regulation of DNA methyltransferases or demethylases serves as the main contributor, we should expect a general but not specific reduction of methylation. With the above data and rational, we think the mis-regulation of DNA methyltransferases and demethylases are not the main reason of reduced DNA methylation in spt6l. It is of note that we found the expression changes of DRM1 (AT5G15380) and DRM2 (AT5G14620) in RNA-seq (Figure 3E) are inconsistent to that of original qPCR. By checking the primers (original Table1 Primer), we noticed that the original qPCR results were generated by wrong primers, which targeted to DRM1 (AT1G28330) and DRM2 (AT2G33830). To confirm the RNA-seq results, we redesigned primers targeted to DRM1 (AT5G15380), DRM2 (AT5G14620) and found similar changes to that in RNA-seq data.

New concerns/comments: In the original version of this manuscript, the Met1 accumulation is noticeably lower in *spt6l* than WT (Suppl. Fig. 4A). But the revised version suggests that there is no statistically different accumulation of Met1 in *spt6l* vs WT (Fig. 3E). Please discuss the differences observed by the two different experiments. Since the revised manuscript no longer has a corresponding panel, all qPCR primers should be removed from Supplementary Table 1. The wrong sets of DRM1/2 primers are still there, also the MET1 primers target the wrong mRNA.

Thank you for your comments. Indeed, the original *MET1* primers target to wrong gene (*AT1G55480*, *ZKT*). And, we designed new pair of primers and performed qPCR at below. The qPCR result indicated that the expression of *MET1* (*AT5G49160*) in *spt6l* is increased more than 2-fold. From the RNA-seq data, the expression of *MET1* in *spt6l* is upregulated 1.69-fold ($\log_2(1.69) = 0.76$, $\text{adj}P = 3.2 \times 10^{-5}$, Datasets 4). With the 2-fold change threshold, we did not highlight the change of *MET1* in Figure 3E. Regardless the specific fold change of *MET1*, both qPCR and RNA-seq data indicated that the reduction of DNA methylation in *spt6l* mutant was not mainly caused by the down-regulation of DNA methyltransferases. In addition, we have removed qPCR primers from Supplementary Table 1.

Our previous review comments: The authors used yeast two-hybrid (supplementary figure 2F/6C) assays to probe interactions between *SPT6L* and Pol V-specific subunits/*SPT5L*, but the results overall are not clear, as *NRPE4*, 7 and *SPT5L* signals are inconsistent among replicates. *NRPE3B* (presumably *AT2G15400*? -this should be clarified) was detected in Pol II by mass spec, as well (Ream TS. et al. 2015 [pmid 25813043]). The Y2H experiments lack proper controls needed to conclude a negative result (ie. Is the *NRPE1* AD construct working properly?).

Authors' response:

Thank you for your comments. We are sorry for missing the important reference you mentioned and we have removed the *NRPE3B* related results from the revised manuscript. For the Y2H results, we have repeated the experiments with serial dilution and provided corresponding controls in Supplementary figure 2F and 6C.

New concerns/comments: The new set of Y2H data (Fig. S2F) shows *NRPE4*-*SPT6L* interaction clearly, but *NRPE7*-*SPT6L* is still not clear. In response to reviewer 1, authors stated that they will not claim interaction between *NRPE7*-*SPT6L*. However, it is still implied in the revised 1.148.

The revised Fig. S2F still does not have a proper control to claim that there is no *NRPE1*-*SPT6L* interaction (1.147). Whereas it is true that no interaction was observed in the experiments, a positive control is needed here, for example the *NRPE1*-*NRPE2* pair, to make sure that the *NRPE1*-AD construct can detect a known positive interaction signal. Without-3AT panels do not

serve this purpose. If there is no interaction detected, the author should at least do the western blot to show that NRPE1 is well expressed in yeast cells.

If NRPE4 is the major interaction surface with SPT6L, it raises a question about Pol IV-SPT6L interaction, as that subunit is also shared with Pol IV. Does SPT6L associate with Pol IV as well?

Thank you for your comments. We have rephased the sentence in the text and provide immunoblotting to confirm the successful expression of different subunits of Pol V in Supplementary Figure 2G.

Indeed, the interaction between SPT6L and NRPE4 raises the potential interaction between Pol IV-SPT6L. Based on the current data, even if SPT6L can interact with Pol IV, we think SPT6L has little effect on Pol IV function and Pol IV is not involved in the recruitment of SPT6L at intergenic loci. The reasons are listed as follow: 1, the mutation of SPT6L has little effect on the proportion and amount of 24 nt siRNA (Figure 3F and 3G), which is mainly generated by Pol IV, indicating that SPT6L is dispensable for the Pol IV transcription. 2, Only two of the six loci showed reduced SPT6L occupancy in the *nrpd1* mutant (Figure 5F), suggesting Pol IV may not directly recruit SPT6L to most of intergenic regions. 3, The mutation of NRPE1 or SPT5L dramatically abolished the intergenic occupancy of SPT6L (Figure 2C and 5B), suggesting that Pol IV has little effect to determine the intergenic occupancy of SPT6L.

Our previous review comments: *To support the claim that SPT6L is involved in Pol V transcription elongation, measurements of full-length Pol V transcripts would be needed. Accumulating evidence indicates that Pol V transcripts are sliced by AGO4. For the RIP-Seq experiment, it is unclear if the Pol V-bound RNAs are sliced or not. To study the size of full-length Pol V transcripts, the authors should do the RIP experiments in ago4 or ago4/6/9 mutant backgrounds.*

Authors' response:

Thank you for your comments. To our knowledge, it is still not clear whether AGO4/6/9 can slice Pol V transcript or not. The result from Wierzbicki's lab (Masayuki Tsuzuki, 2020) indicates that the Pol V transcript is general unchanged in ago4 mutant. Their following up work (M. Hafiz Rothi, 2021) further divided the Pol V transcripts into AGO4 dependent (I) and independent (II) and concluded that the different levels of DNA methylation between (I) and (II) linked to Pol V transcripts. This result suggests that the slicing role of AGO4 to Pol V transcript, if it exists, will not function equally to all the Pol V transcripts. In our case, the reduced Pol V transcripts has been found within most of Pol V peaks (Figure 6C).

The most direct evidence for the slicing role of AGO4 on Pol V transcripts, to our knowledge, is the reduced GRO-seq signals in ago4 by comparing the GRO-seq signals from ago4 nrpe1, nrpe1, and WT (Liu, 2018). As we mentioned in discussion part, this work prepared their GRO-seq library by exclusively selecting 5' monophosphorylated RNA, which is about 30% of total Pol V transcripts (estimated by Pikaard's lab, Wendte, 2017). Thus, it seems the slicing role of AGO4 may be only effective on a small proportion of Pol V transcripts.

At last, according to current RdDM model, siRNA is required for AGO4 to associated with Pol V/Pol V transcripts. We found the level of siRNA is comparable between WT and spt6l (Figure 3F

and 3G). And, in AGO4 side, the newly provided RNAseq data indicated that the expression of AGO4 in WT is similar to that in *spt6l* (Figure 3E), suggesting that the effect of AGO4-slice, if it exists, will be similar in WT and *spt6l*. Thus, the reduced Pol V transcripts and read length in *spt6l* unlikely result from changed AGO4 slicing ability.

Altogether, we think the examination of the Pol V transcripts in *ago4* or even *ago4/6/9* mutant will be interesting, but it will not affect the result of SPT6L's role in Pol V transcription.

New concerns/comments: A recent publication (Wang et al., Genes & Development, 2023) shows that AGO4 has slicer activity both in vivo and in vitro. This paper clearly showed that the slicer activity of AGO4 is needed at virtually all RdDM target loci.

The average lengths of Pol V transcripts described in Fig. 6D RIP-seq are noticeably shorter compared to what have been reported previously (ref 18 in the text l.67). It is possible that the method used in this manuscript did not capture full-length Pol V transcripts, regardless of AGO4 slicing. Transcription elongation is a specific step in the transcription cycle of a Pol (i.e. initiation, elongation, and termination). While SPT6L's involvement in Pol V function in general can be discussed, claiming that SPT6L mediates Pol V transcription elongation, based on these data, is a gross over-interpretation. In the title, and throughout the manuscript, I suggest changing "Pol V elongation" to "Pol V transcription", as the data presented in this study does not provide direct observations of Pol V transcription elongation nor specific examination of full-length Pol V transcripts.

Thank you for your comments. We agree that current manuscript does not provide direct evidence to support the regulation of SPT6L on Pol V elongation. Thus, we have changed our title and text as reviewer suggested.

We have gone through the work mentioned by the reviewer. After comparing the *ago4* DMR regions and AGO4-bound RNAs with Pol V peaks (7,809 peaks), we found only half or two-of-third Pol V peaks were overlapped, respectively. And then, we compared the Pol V GRO-seq data to Pol V peaks and found more than 7,300 Pol V peaks contained the GRO-seq reads. With those reads, we successfully captured the slicing feature (Supplementary Figure 7A). Together, we agree that the slicing events likely happened in all the Pol V bound regions. And, as shown in Figure 6E, the Pol V transcripts generated from different regions were sliced with vary preferences. These preferences can also be detected in both NRPE1 and *spt6l* NRPE1 RIP-seq data (Figure 6E). Thus, we think the mutation of SPT6L may not affect the slicing ability of AGOs.

Our previous review comments: *The genomic data is difficult to interpret. Even though the authors presented Pol V, Pol II, and SPT6L ChIP-seq results in the manuscript, it is unclear how many SPT6L peaks overlap with Pol II and Pol IV. Dissecting the genome into 6 groups (S1 to S6 in the manuscript) does not help the reader to understand the interaction sites of SPT6L. A more defined classification of genomic regions, such as genic regions, intergenic regions, and TEs, etc., would be more useful. "TSS" in this paper's context is also confusing, as it represents Pol II's TSS sites, not Pol V's.*

Authors' response:

Thank you for your comments. Originally, we have provided the peak lists in Supplementary Dataset1, which includes SPT6L genic peak (associated with Pol II), SPT6L intergenic peaks (associated with Pol V), NRPE1 and SPT6L overlapped peaks, NRPE1-only peaks. For the overlapping with different genomic feature, we have provided the proportions of TE, gene, and others in Supplementary Figure 1B. To the divided genomic states (6 states), we have provided the enrichment of each state on different genomic features in Supplementary Figure 2E. Plus, in this revision, we further analyzed the categories of TE associated by NRPE1 and NRPE1-SPT6L peaks (Supplementary Figure 1E). For the word "TSS", indeed, we referred to Pol II transcription starting sites rather than Pol V. The TSS was mainly mentioned in Figure 2D. In this part, we were analyzed the change of Pol II occupancy at the nearest downstream genes of Pol V peak. It was reported that Pol V can affect the expression of downstream genes and the affection was getting obvious with the closing distance between Pol V peak and downstream TSS (Zhong, X. et al, 2012). To make this point clear, we have rephrased the words and sentences.

New concerns/comments: It is still unclear what, exactly, the 6 genomic states are. Please define the 6 genomic states in simple words, such as Pol V ChIP peaks, TE regions, or upstream of TSS, etc.

Thank you for your comments. We understand the simple name of region will be easy for readers to recall the feature. But, from current data, we can not define simple words to accurately replaces the genomic states. For example, Pol V peaks equals to states G2 and G3, these two parts were separated based on whether or not SPT6L were enriched. We may change G2 and G3 to Pol V-only and Pol V-SPT6L shared, but the problem is still there (what do the Pol V-SPT6L shared peaks represent?). In fact, the Pol V peak, which enriched with SPT6L signals, has relative high level of DNA methylation and TE content compared with Pol V only peak. But it doesn't mean Pol V only peaks do not contain TEs or DNA methylation. We have tried to compare the DNA methylation changes between G2 and G3 in different RdDM mutants, however, we always found a similar change between them. Above all, we would like to keep the name of genomic state as it is.

Specific comments

Our previous review comments: l.61: *"The short products from Pol IV and V are likely due to the lack of surfaces to recruit Pol II transcription factors such as TFIIB and TFIIS"*This statement is misleading and at odds with what is known. Evidence indicates that Pol IV transcripts are short because of Pol IV's inability to elongate through double-stranded DNA. Pol V transcript length has only been estimated, not demonstrated empirically. But RT-PCR and other analyses by Wierzbicki and colleagues have suggested that Pol V transcripts may be 200 nt or longer.

Authors' response:

Thank you for your comments. Indeed, our original description is overstatement. We have rephrased the text and introduced the previous in vitro data about the biochemical nature of Pol V. The recent structural data just provide further explanation why Pol IV and V can not transcript smoothly.

New concerns/comments: 1.67-76: The newly added statements are still misleading. Recent structural studies have provided possibilities for Pol V being prone to backtracking, like Pol IV. This may help explain why Pol V transcription shows higher fidelity than Pol IV, because backtracking is a part of the proofreading process used by multi-subunit Pols. However, this does not explain why Pol V can produce RNAs significantly longer than Pol IV. The notion of "Pol V being inefficient at transcription" is pure speculation due to the lack of studies of productive Pol V transcription elongation in the field. It should not be stated as a fact.

Thank you for your comments. We agree the current data did not directly support the notion of the inefficiency of Pol V. Thus, we changed the sentence to emphasize the distinct regulation of Pol II and Pol V in transcription.

1.101: "transposon elements"  "transposable elements"

1.105-106: "large subunits"  "largest subunit"

1.317: "significant"  "significantly"

Thank you for your comments and we have fixed the above typos.

1.322-323: "reduced occupancies of SPT6L in nrpd1"

Based on Figure 5F, only 2 of the 6 loci shows SPT6L occupancy being reduced in the nrpd1 mutant.

Thank you for your comment. The claim in 1.322-323 refers the 2 loci but not a general all the SPT6L binding sites. We have rephrased the words to avoid the misunderstanding.

Figure 1E and 5D: Specify which antibody you are using to do the IP in the legend

Thank you for your comment and we have added the related information in the Figure legend.

REVIEWERS' COMMENTS

Reviewer #2 (Remarks to the Author):

The authors have addressed most of our concerns. They revised the discussion on SPT6L's role in Pol V transcription, toning down their claims to a more appropriate level. Based on new analyses, the authors propose that SPT6L is not directly involved in Pol V transcript slicing by AGO4. The MET1 and qPCR-related concerns have been addressed by the new qPCR with correct primers and removing the previous erroneous information.

Before publication, a few corrections are still suggested/needed:

- The nomenclature of Pol V subunits should reflect whether they are shared by Pol IV and/or Pol II. Shared subunit NRPE2 should be written as NRP(D/E)2, likewise NRPE4 should be written as NRP(D/E)4, throughout the manuscript.
- The authors' response clarifies the reasons why SPT6L is likely dispensable for Pol IV function despite NRP(D/E)4-specific interaction revealed by Y2H, based on their in vivo observations. It would be useful to add this discussion to the text of the paper where the spt6l mutant effects on 24-nt siRNA are described (Fig. 3F/G).
- In Figure 3E, the label "DDR complex" incorrectly indicates DDM1 as a part of the complex along with DMS3 and RDM1. The "DDR" means DRD1-DMS3-RDM1, not DDM1. The "DDR complex" label should be removed, or the DDM1 data should be replaced with the correct name for DRD1 (AT2G16390).
- In Supplemental Figure 7B, do the logos ranging from "-20" to "0" represent reference DNA sequences? If this is the scenario, please change U to T and label this region as genomic DNA.

The authors have addressed most of our concerns. They revised the discussion on SPT6L's role in Pol V transcription, toning down their claims to a more appropriate level. Based on new analyses, the authors propose that SPT6L is not directly involved in Pol V transcript slicing by AGO4. The MET1 and qPCR-related concerns have been addressed by the new qPCR with correct primers and removing the previous erroneous information.

Before publication, a few corrections are still suggested/needed:

- The nomenclature of Pol V subunits should reflect whether they are shared by Pol IV and/or Pol II. Shared subunit NRPE2 should be written as NRP(D/E)2, likewise NRPE4 should be written as NRP(D/E)4, throughout the manuscript.

Thank you for the correction. All the shared subunits were revised in manuscript.

- The authors' response clarifies the reasons why SPT6L is likely dispensable for Pol IV function despite NRP(D/E)4-specific interaction revealed by Y2H, based on their in vivo observations. It would be useful to add this discussion to the text of the paper where the spt6l mutant effects on 24-nt siRNA are described (Fig. 3F/G).

Thank you for your suggestions and we already add this part in discussion part.

- In Figure 3E, the label "DDR complex" incorrectly indicates DDM1 as a part of the complex along with DMS3 and RDM1. The "DDR" means DRD1-DMS3-RDM1, not DDM1. The "DDR complex" label should be removed, or the DDM1 data should be replaced with the correct name for DRD1 (AT2G16390).

Thank you for the correction. We already replaced the DDM1 to DRD1 in Figure 3E.

- In Supplemental Figure 7B, do the logos ranging from "-20" to "0" represent reference DNA sequences? If this is the scenario, please change U to T and label this region as genomic DNA.

Thank you for point out the incorrect labeling and we already changed length of RNA from +1 to +25 and only show U rather than T in the sequence logo.